# LINEAR INDEXED MINIMUM EMPIRICAL DIVERGENCE ALGORITHMS

## ABSTRACT

The Indexed Minimum Empirical Divergence (IMED) algorithm is a highly effective approach that offers a stronger theoretical guarantee of the asymptotic optimality compared to the Kullback–Leibler Upper Confidence Bound (KL-UCB) algorithm for the multi-armed bandit problem. Additionally, it has been observed to empirically outperform UCB-based algorithms and Thompson Sampling. Despite its effectiveness, the generalization of this algorithm to contextual bandits with linear payoffs has remained elusive. In this paper, we present novel linear versions of the IMED algorithm, which we call the family of LinIMED algorithms. We demonstrate that LinIMED provides a $\widetilde{O}(d\sqrt{T})$ upper regret bound where $d$ is the dimension of the context and $T$ is the time horizon. Furthermore, empirical studies reveal that LinIMED and its variants outperform widely-used linear bandit algorithms such as LinUCB and Linear Thompson Sampling in some regimes.

## 1 INTRODUCTION

The multi-armed bandit (MAB) problem (Lattimore & Szepesvári (2020)) is a classical topic in decision theory and reinforcement learning. Among the various subfields of bandit problems, the stochastic linear bandit is the most popular area due to its wide applicability in large-scale, real-world applications such as personalized recommendation systems (Li et al. (2010)), online advertising, and clinical trials. In the stochastic linear bandit model, at each time step $t$, the learner has to choose one arm $A_t$ from the time-varying action set $\mathcal{A}_t$. Each arm $a \in \mathcal{A}_t$ has a corresponding context $x_{t,a} \in \mathbb{R}^d$, which is a $d$-dimensional vector. By pulling the arm $a \in \mathcal{A}_t$ at time step $t$, under the linear bandit setting, the learner will receive the reward $Y_{t,a}$, whose expected value satisfies $\mathbb{E}[Y_{t,a}|x_{t,a}] = \langle \theta^*, x_{t,a} \rangle$, where $\theta^* \in \mathbb{R}^d$ is an unknown parameter. The goal of the learner is to maximize his cumulative reward over a time horizon $T$, which also means minimizing the cumulative regret, defined as $R_T := \mathbb{E}\left[\sum_{t=1}^{T} \max_{a \in \mathcal{A}_t} Y_{t,a} - Y_{t,A_t}\right]$. The learner needs to balance the trade-off between the exploration of different arms (to learn their expected rewards) and the exploitation of the arm with the highest expected reward based on the available data. When the action set $\mathcal{A}_t$ is varying over time but finite (i.e., $|\mathcal{A}_t| < \infty$), we term this setting as the stochastic linear bandit with finite but varying arm setting, and this is the setting this paper focused on.

### 1.1 MOTIVATION AND RELATED WORK

The $K$-armed bandit setting is a special case of the linear bandit. There exist several good algorithms such as UCB1 (Auer et al. (2002)), Thompson Sampling (Agrawal & Goyal (2012)), and the Indexed Minimum Empirical Divergence (IMED) algorithm (Honda & Takemura (2015)) for this setting. There are three main families of asymptotically optimal multi-armed bandit algorithms based on different principles (Baudry et al. (2023)). However, among these algorithms, only IMED lacks an extension for contextual bandits with linear payoff. In the context of the varying arm setting of the linear bandit problem, the LinUCB algorithm in Li et al. (2010) is frequently employed in practice. It has a theoretical guarantee on the regret in the order of $O(d\sqrt{T}\log(T))$ using the confidence width as in OFUL (Abbasi-Yadkori et al. (2011)). Although the SupLinUCB algorithm introduced by Chu et al. (2011) uses phases to decompose the reward dependence of each time step and achieves an $\widetilde{O}(\sqrt{dT})$ (the $\widetilde{O}(\cdot)$ notation omits logarithmic factors in $T$) regret upper bound, its empirical performance falls short of both the algorithm in Li et al. (2010) and the Linear Thompson Sampling algorithm (Agrawal & Goyal (2013)) as mentioned in Lattimore & Szepesvári (2020, Chapter 22).

| | Problem indepen-dent regret bound | Efficient for large finite arm sets? | Regret bound in-dependent of $K$? | Principle that the algo-rithm is based on |
|---|---|---|---|---|
| OFUL (Abbasi-Yadkori et al. (2011)) | $O(d\sqrt{T}\log(T))$ | ✗ | ✓ | Optimism |
| LinUCB (Li et al. (2010)) | Difficult to analyze | Not Applicable | Unknown | Optimism |
| LinTS (Agrawal & Goyal (2013)) | $O(d^{\frac{3}{2}}\sqrt{T}) \quad \wedge$ $O(d\sqrt{T\log(K)})$ | ✓ | ✓ | Posterior sampling |
| SupLinUCB (Chu et al. (2011)) | $O(\sqrt{dT\log^3(KT)})$ | ✓ | ✗ | Optimism |
| LinUCB with OFUL's confidence bound | $O(d\sqrt{T}\log(T))$ | ✓ | ✓ | Optimism |
| LinIMED-3 (this paper) | $O(d\sqrt{T}\log(T))$ | ✓ | ✓ | Min. empirical divergence |

Table 1: Comparison of algorithms for linear bandits with finite, varying arm sets

On the other hand, the Optimism in the Face of Uncertainty Linear (OFUL) bandit algorithm in Abbasi-Yadkori et al. (2011) achieves a regret upper bound of $\widetilde{O}(d\sqrt{T})$ through an improved analysis of the confidence bound using a martingale technique. However, it involves a bilinear optimization problem over the action set and the confidence ellipsoid when choosing the arm at each time. This is computationally expensive, unless the confidence ellipsoid is a convex hull of a finite set.

For randomized algorithms designed for the linear bandit problem, Agrawal & Goyal (2013) proposed the LinTS algorithm, which is in the spirit of Thompson Sampling (Thompson (1933)) and the confidence ellipsoid similar to that of LinUCB-like algorithms. This algorithm performs efficiently and achieves a regret upper bound of $O(d^{\frac{3}{2}}\sqrt{T} \wedge d\sqrt{T\log K})$, where $K$ is the number of arms at each time step such that $|\mathcal{A}_t| = K$ for all $t$. Compared to LinUCB with OFUL's confidence width, it has an extra $O(\sqrt{d} \wedge \sqrt{\log K})$ term for the minimax regret upper bound.

Recently, MED-like (minimum empirical divergence) algorithms have come to the fore since these randomized algorithms have the property that the probability of selecting each arm is in closed form, which benefits downstream work such as offline evaluation with the inverse propensity score. Both MED in the sub-Gaussian environment and its deterministic version IMED have demonstrated superior performances over Thompson Sampling (Bian & Jun (2021), Honda & Takemura (2015)). Baudry et al. (2023) also shows MED has a strong relationship with Thompson Sampling. In particular, it is argued that MED and TS can be interpreted as two variants of the same exploration strategy. Bian & Jun (2021) also shows that probability of selecting each arm of MED in the sub-Gaussian case can be viewed as a closed-form approximation of the same probability as in Thompson Sampling. We take inspiration from the extension of Thompson Sampling to linear bandits and thus are motivated to extend MED-like algorithms to the linear bandit setting and prove regret bounds that are competitive vis-à-vis the state-of-the-art bounds.

Thus, this paper aims to answer the question of whether it is possible to devise an extension of the IMED algorithm for the linear bandit problem in the finite and varying arm setting with a regret upper bound of $O(d\sqrt{T}\log T)$ which matches LinUCB with OFUL's confidence bound while being as efficient as LinUCB. The proposed algorithm, called LinIMED, can be viewed as a generalization of the IMED algorithm (Honda & Takemura (2015)) to the linear bandit setting. We prove that LinIMED and its variants achieve a regret upper bound of $\widetilde{O}(d\sqrt{T})$ and they perform efficiently, no worse than LinUCB. Moreover, in our empirical study, we found that the different variants of LinIMED perform even better than LinUCB and LinTS for the synthetic and real-world instances under consideration.

Compared to OFUL, LinIMED works more efficiently. Compared to SupLinUCB, our algorithm is significantly simpler, and compared to LinUCB with OFUL's confidence bound, our empirical performance is better. This is because in our algorithm, the exploitation term and exploration term are independent and this leads to a finer control while tuning the hyperparameters in the empirical study.

Finally, compared to LinTS, our algorithm's (specifically LinIMED-3) regret bound is superior, by an order of $O(\sqrt{d} \wedge \sqrt{\log K})$. Since fixed arm setting is a special case of finite varying arm setting, our result is more general than other fixed-arm linear bandit algorithms like Spectral Eliminator (Valko et al. (2014)) and PEGOE (Lattimore & Szepesvári (2020, Chapter 22)). We summarize the comparisons of LinIMED to other linear bandit algorithms in Table 1.

## 2 PROBLEM STATEMENT

**Notations:** For any $d$ dimensional vector $x \in \mathbb{R}^d$ and a $d \times d$ positive definite matrix $A$, we use $\|x\|_A$ to denote the Mahalanobis norm $\sqrt{x^\top A x}$. We use $a \wedge b$ (resp. $a \vee b$) to represent the minimum (resp. maximum) of two real numbers $a$ and $b$.

**The Stochastic Linear Bandit Model:** In the stochastic linear bandit model, the learner chooses an arm $A_t$ at each round $t$ from the arm set $\mathcal{A}_t = \{a_{t,1}, a_{t,2}, \ldots, a_{t,K}\} \subseteq \mathbb{R}$, where we assume the cardinality of each arm set $\mathcal{A}_t$ is fixed to be a constant such that $|\mathcal{A}_t| = K$ for all $t \geq 1$. Each arm $a \in \mathcal{A}_t$ at time $t$ has a corresponding context $x_{t,a}$, which is known to the learner. After choosing arm $A_t$, the environment reveals the reward

$$Y_t = \langle \theta^*, X_t \rangle + \eta_t$$

to the learner where $X_t := x_{t,A_t}$ is the corresponding context of the arm $A_t$, $\theta^* \in \mathbb{R}^d$ is an unknown coefficient of the linear model, $\eta_t$ is an $R$-sub-Gaussian noise conditioned on $\{A_1, A_2, \ldots, A_t, Y_1, Y_2, \ldots, Y_{t-1}\}$ such that for any $\lambda \in \mathbb{R}$, almost surely,

$$\mathbb{E}\left[\exp(\lambda \eta_t) \mid A_1, A_2, \ldots, A_t, Y_1, Y_2, \ldots, Y_{t-1}\right] \leq \exp\left(\frac{\lambda^2 R^2}{2}\right).$$

Denote $a_t^* := \arg\max_{a \in \mathcal{A}_t} \langle \theta^*, x_{t,a} \rangle$ as the arm with the largest reward at time $t$. The goal of the learner is to minimize the expected cumulative regret over the horizon $T$. The (expected) cumulative regret is defined as

$$R_T = \mathbb{E}\left[\sum_{t=1}^T \langle \theta^*, x_{t,a_t^*} \rangle - \langle \theta^*, X_t \rangle\right].$$

**Assumption 1.** *For each time $t$, we assume that $\|X_t\| \leq L$, and $\|\theta^*\| \leq S$ for some fixed $L, S > 0$. We also assume that $\Delta_{t,b} := \max_{a \in \mathcal{A}_t} \langle \theta^*, x_{t,a} \rangle - \langle \theta^*, x_{t,b} \rangle \leq 1$ for each arm $b \in \mathcal{A}_t$ and time $t$.*

## 3 DESCRIPTION OF LINIMED ALGORITHMS

In the pseudocode of Algorithm 1, for each time step $t$, in Line 4, we use the improved confidence bound of $\theta^*$ as in Abbasi-Yadkori et al. (2011) to calculate the confidence bound $\beta_{t-1}(\gamma)$. After that, for each arm $a \in \mathcal{A}_t$, in Lines 7 to 8, the empirical gap between the highest empirical reward and the empirical reward of arm $a$ is estimated as

$$\hat{\Delta}_{t,a} = \begin{cases} \max_{j \in \mathcal{A}_t} \langle \hat{\theta}_{t-1}, x_{t,j} \rangle - \langle \hat{\theta}_{t-1}, x_{t,a} \rangle & \text{if LinIMED-1,2} \\ \max_{j \in \mathcal{A}_t} \text{UCB}_t(j) - \text{UCB}_t(a) & \text{if LinIMED-3} \end{cases}$$

Then, in Lines 10 to 12, with the use of the confidence width of $\beta_{t-1}(\gamma)$, we can compute the index $I_{t,a}$ for the empirical best arm $a = \arg\max_{j \in \mathcal{A}_t} \hat{\mu}_{t,a}$ (for LinIMED-1,2) or the highest UCB arm $a = \arg\max_{j \in \mathcal{A}_t} \text{UCB}_j(a)$ (for LinIMED-3). The different versions of LinIMED encourage different amounts of exploitation. For the other arms, in Line 14, the index is defined and computed as

$$I_{t,a} = \frac{\hat{\Delta}_{t,a}^2}{\beta_{t-1}(\gamma)\|x_{t,a}\|_{V_{t-1}^{-1}}^2} + \log\frac{1}{\beta_{t-1}(\gamma)\|x_{t,a}\|_{V_{t-1}^{-1}}^2}.$$

Then with all the indices of the arms calculated, in Line 17, we choose the arm $A_t$ with the minimum index such that $A_t = \arg\min_{a \in \mathcal{A}_t} I_{t,a}$ (where ties are broken arbitrarily) and the agent receives its reward. Finally, in Lines 19 to 21, we use ridge regression to estimate the unknown $\theta^*$ as $\hat{\theta}_t$ and update the matrix $V_t$ and the vector $W_t$. After that, the algorithm iterates to the next time step until the time horizon $T$. From the pseudo-code, we observe that the only differences between the three algorithms are the way that the square gap, which plays the role of the empirical divergence, is estimated and the index of the empirically best arm. The latter point implies that we encourage the empirically best arm to be selected more often in LinIMED-2 and LinIMED-3 compared to LinIMED-1; in other words, we encourage more exploitation in LinIMED-2 and LinIMED-3. Similar to the core spirit of IMED algorithm Honda & Takemura (2015), the first term of our index $I_{t,a}$ for LinIMED-1 algorithm is $\hat{\Delta}_{t,a}^2/(\beta_{t-1}(\gamma)\|x_{t,a}\|_{V_{t-1}^{-1}}^2)$, this is the term controls the exploitation, while the second term $-\log(\beta_{t-1}(\gamma)\|x_{t,a}\|_{V_{t-1}^{-1}}^2)$ controls the exploration in our algorithm.

---

**Algorithm 1** LinIMED-$x$ for $x \in \{1, 2, 3\}$

---

1: **Input:** Dimension $d$, Regularization parameter $\lambda$, Bound $S$ on $\|\theta^*\|$, Sub-Gaussian parameter $R$, Concentration parameter $\gamma$ of $\theta^*$, Bound $L$ on $\|x_{t,a}\|$ for all $t \geq 1$ and $a \in \mathcal{A}_t$, Constant $C \geq 1$.

2: **Initialize:** $V_0 = \lambda I_{d \times d}$, $W_0 = 0_{d \times 1}$(all zeros vector with $d$ dimensions), $\hat{\theta}_0 = V_0^{-1} W_0$

3: **for** $t = 1, 2, \ldots T$ **do**

4:     Receive the arm set $\mathcal{A}_t$ and compute $\beta_{t-1}(\gamma) = (R\sqrt{d \log(\frac{1+(t-1)L^2/\lambda}{\gamma})} + \sqrt{\lambda}S)^2$.

5:     **for** $a \in \mathcal{A}_t$ **do**

6:         Compute:

7:         $\hat{\mu}_{t,a} = \langle \hat{\theta}_{t-1}, x_{t,a} \rangle$ and $\hat{\Delta}_{t,a} = \max_{j \in \mathcal{A}_t} \hat{\mu}_{t,j} - \hat{\mu}_{t,a}$ (LinIMED-1, 2)

8:         $\text{UCB}_t(a) = \langle \hat{\theta}_{t-1}, x_{t,a} \rangle + \sqrt{\beta_{t-1}(\gamma)}\|x_{t,a}\|_{V_{t-1}^{-1}}$ and

            $\hat{\Delta}_{t,a} = \max_{j \in \mathcal{A}_t} \text{UCB}_t(j) - \text{UCB}_t(a)$ (LinIMED-3)

9:         **if** $a = \arg\max_{j \in \mathcal{A}_t} \hat{\mu}_{t,a}$ (LinIMED-1,2) or $a = \arg\max_{j \in \mathcal{A}_t} \text{UCB}_t(a)$ (LinIMED-3)
            **then**

10:             $I_{t,a} = -\log(\beta_{t-1}(\gamma)\|x_{t,a}\|_{V_{t-1}^{-1}}^2)$            (LinIMED-1)

11:             $I_{t,a} = \log T \wedge (-\log(\beta_{t-1}(\gamma)\|x_{t,a}\|_{V_{t-1}^{-1}}^2))$    (LinIMED-2)

12:             $I_{t,a} = \log \frac{C}{\max_{a \in \mathcal{A}_t} \hat{\Delta}_{t,a}^2} \wedge (-\log(\beta_{t-1}(\gamma)\|x_{t,a}\|_{V_{t-1}^{-1}}^2))$   (LinIMED-3)

13:         **else**

14:             $I_{t,a} = \frac{\hat{\Delta}_{t,a}^2}{\beta_{t-1}(\gamma)\|x_{t,a}\|_{V_{t-1}^{-1}}^2} - \log(\beta_{t-1}(\gamma)\|x_{t,a}\|_{V_{t-1}^{-1}}^2)$

15:         **end if**

16:     **end for**

17:     Pull the arm $A_t = \arg\min_{a \in \mathcal{A}_t} I_{t,a}$ (ties are broken arbitrarily) and receive its reward $Y_t$.

18:     **Update:**

19:     $V_t = V_{t-1} + X_t X_t^\top$

20:     $W_t = W_{t-1} + Y_t X_t$

21:     $\hat{\theta}_t = V_t^{-1} W_t$

22: **end for**

---

### 3.1 RELATION TO THE IMED ALGORITHM OF HONDA & TAKEMURA (2015)

The IMED algorithm is a deterministic algorithm for the $K$-armed bandit problem. At each time step $t$, it chooses the arm $a$ with the minimum index

$$a = \arg\min_{i \in [K]} T_i(t) D_{\inf}(\hat{F}_i(t), \hat{\mu}^*(t)) + \log T_i(t), \tag{1}$$

where $T_i(t) = \sum_{s=1}^{t-1} \mathbb{1}\{A_t = a\}$ is the total arm pulls of the arm $i$ until time $t$ and $D_{\inf}(\hat{F}_i(t), \hat{\mu}^*(t))$ is some divergence measure between the empirical distribution of the sample mean for arm $i$ and the arm with the highest sample mean. More precisely, $D_{\inf}(F, \mu) := \inf_{G \in \mathcal{G}:\mathbb{E}(G) \leq \mu} D(F\|G)$ and $\mathcal{G}$ is the family of distributions supported on $(-\infty, 1]$. As shown in Honda & Takemura (2015), its asymptotic bound is even better than KL-UCB (Garivier & Cappé (2011)) algorithm and can be extended to semi-bounded support models such as $\mathcal{G}$. Also, this algorithm empirically outperforms the Thompson Sampling algorithm as shown in Honda & Takemura (2015). However, the linear extension of IMED algorithm was, prior to our work, still unknown. In our design of LinIMED algorithm, we replace the optimized KL-divergence measure in IMED in Eqn. (1) with the squared gap between the sample mean of the arm $i$ and the arm with the maximum sample mean. This choice simplifies our analysis and does not adversely affect the regret bound. On the other hand, we view the term $1/T_i(t)$ as the variance of the sample mean of arm $i$ at time $t$; then in this spirit, we use $\beta_{t-1}(\gamma)\|x_{t,a}\|_{V_{t-1}^{-1}}^2$ as the variance of the sample mean (which is $\langle \hat{\theta}_{t-1}, x_{t,a} \rangle$) of arm $a$ at time $t$.

## 4 THEOREM STATEMENTS

**Theorem 1.** *Under Assumption 1, the assumption that $\langle \theta^*, x_{t,a} \rangle \geq 0$ for all $t \geq 1$ and $a \in \mathcal{A}_t$, and the assumption that $\sqrt{\lambda}S \geq 1$, the regret of the LinIMED-1 algorithm is upper bounded as follows:*

$$R_T \leq O\big(d\sqrt{T}\log^{\frac{3}{2}}(T)\big).$$

A proof sketch of Theorem 1 is provided in Section 5.

**Theorem 2.** *Under Assumption 1, and the assumption that $\sqrt{\lambda}S \geq 1$, the regret of the LinIMED-2 algorithm is upper bounded as follows:*

$$R_T \leq O\left(d\sqrt{T}\log^{\frac{3}{2}}(T)\right).$$

**Theorem 3.** *Under Assumption 1, the assumption that $\sqrt{\lambda}S \geq 1$, and that $C$ in Line 12 is a constant, the regret of the LinIMED-3 algorithm is upper bounded as follows:*

$$R_T \leq O\left(d\sqrt{T}\log(T)\right).$$

The upper bounds on the regret of LinIMED and its variants are all of the form $\widetilde{O}(d\sqrt{T})$, which, ignoring the logarithmic term, is the same as OFUL algorithm (Abbasi-Yadkori et al. (2011)). The bounds have a gap of $\sqrt{d}$ compared to the lower bound in Chu et al. (2011). Compared to LinTS, it has an advantage of $O(\sqrt{d} \wedge \sqrt{\log K})$. Also, these upper bounds do not dependent on the number of arms $K$, which means it can be applied to linear bandit problems with a large but finite arm set. One observes that LinIMED-2 and LinIMED-3 do not require the additional assumption that $\langle \theta^*, x_{t,a} \rangle \geq 0$ for all $t \geq 1$ and $a \in \mathcal{A}_t$ to achieve the $\widetilde{O}(d\sqrt{T})$ upper regret bound. It is difficult to prove the regret bound for the LinIMED-1 algorithm without this assumption since in our proof we need to use that $\langle \theta^*, X_t \rangle \geq 0$ for any time $t$ to bound the $F_1$ term. On the other hand, LinIMED-2 and LinIMED-3 encourage more exploitations in terms of the index of the empirically best arm at each time without adversely influencing the regret bound; this will accelerate the learning with well-preprocessed datasets. The regret bound of LinIMED-3, in fact, matches that of LinUCB with OFUL's confidence bound. In the proof, we will extensively use a technique known as the "peeling device" (Lattimore & Szepesvári, 2020, Chapter 9). This analytical technique, commonly used in the theory of bandit algorithms, involves the partitioning of the range of some random variable into several pieces, then using the basic fact that $\mathbb{P}(A \cap (\cup_{i=1}^{\infty} B_i)) \leq \sum_{i=1}^{\infty} \mathbb{P}(A \cap B_i)$, we can utilize the more *refined range* of the random variable to derive desired bounds.

## 5   PROOF SKETCH OF THEOREM 1

We choose to present the proof sketch of Theorem 1 since it contains the main ingredients. Before presenting the proof, we introduce the following lemma and corollary.

**Lemma 1.** *(Abbasi-Yadkori et al. (2011, Theorem 2)) Under Assumption 1, for any time step $t \geq 1$ and any $\gamma > 0$, we have*

$$\mathbb{P}\left(\|\hat{\theta}_{t-1} - \theta^*\|_{V_{t-1}} \leq \sqrt{\beta_{t-1}(\gamma)}\right) \geq 1 - \gamma.$$

This lemma illustrates that the true parameter $\theta^*$ lies in the ellipsoid centered at $\hat{\theta}_{t-1}$ with high probability, which also states the width of the confidence bound.

The second is a corollary of the elliptical potential count lemma in Abbasi-Yadkori et al. (2011):

**Corollary 1.** *(Corollary of Lattimore & Szepesvári (2020, Exercise 19.3)) Assume that $V_0 = \lambda I$ and $\|X_t\| \leq L$ for $t \in [T]$, for any constant $0 < m \leq 2$, the following holds:*

$$\sum_{t=1}^{T} \mathbb{1}\left\{\|X_t\|_{V_{t-1}^{-1}}^2 \geq m\right\} \leq \frac{6d}{m}\log\left(1 + \frac{2L^2}{\lambda m}\right).$$

*Proof.* First we define $a_t^*$ as the best arm in time step $t$ such that $a_t^* = \arg\max_{a \in \mathcal{A}_t}\langle \theta^*, x_{t,a} \rangle$, and use $x_t^* := x_{t,a_t^*}$ denote its corresponding context. Let $\Delta_t := \langle \theta^*, x_t^* \rangle - \langle \theta^*, X_t \rangle$ denote the regret in time $t$. Define the following events:

$$B_t := \left\{\|\hat{\theta}_{t-1} - \theta^*\|_{V_{t-1}} \leq \sqrt{\beta_{t-1}(\gamma)}\right\}, \quad C_t := \left\{\max_{b \in \mathcal{A}_t}\langle \hat{\theta}_{t-1}, x_{t,b} \rangle > \langle \theta^*, x_t^* \rangle - \delta\right\}$$

$$D_t := \left\{\hat{\Delta}_{t,A_t} \geq \varepsilon\right\}.$$

where $\delta$ and $\varepsilon$ are free parameters set to be $\delta = \frac{\Delta_t}{\sqrt{\log T}}$ and $\varepsilon = (1 - \frac{2}{\sqrt{\log T}})\Delta_t$ in this proof sketch.

Then the expected regret $R_T = \mathbb{E}\sum_{t=1}^T \Delta_t$ can be partitioned by events $B_t, C_t, D_t$ such that:

$$R_T = \underbrace{\mathbb{E}\sum_{t=1}^T \Delta_t \cdot \mathbb{1}\{B_t, C_t, D_t\}}_{=:F_1} + \underbrace{\mathbb{E}\sum_{t=1}^T \Delta_t \cdot \mathbb{1}\{B_t, C_t, \overline{D}_t\}}_{=:F_2} + \underbrace{\mathbb{E}\sum_{t=1}^T \Delta_t \cdot \mathbb{1}\{B_t, \overline{C}_t\}}_{=:F_3}$$

$$+ \underbrace{\mathbb{E}\sum_{t=1}^T \Delta_t \cdot \mathbb{1}\{\overline{B}_t\}}_{=:F_4}.$$

For $F_1$, from the event $C_t$ and the fact that $\langle\theta^*, x_t^*\rangle = \Delta_t + \langle\theta^*, X_t\rangle \geq \Delta_t$ (here is where we use that $\langle\theta^*, x_{t,a}\rangle \geq 0$ for all $t$ and $a$), we obtain $\max_{b\in\mathcal{A}_t}\langle\hat{\theta}_{t-1}, x_{t,b}\rangle > (1 - \frac{1}{\sqrt{\log T}})\Delta_t$. For convenience, define $\hat{A}_t := \arg\max_{b\in\mathcal{A}_t}\langle\hat{\theta}_{t-1}, x_{t,b}\rangle$ as the empirically best arm at time step $t$, where ties are broken arbitrarily, then use $\hat{X}_t$ to denote the corresponding context of the arm $\hat{A}_t$. Therefore from the Cauchy–Schwarz inequality, we have $\|\hat{\theta}_{t-1}\|_{V_{t-1}}\|\hat{X}_t\|_{V_{t-1}^{-1}} \geq \langle\hat{\theta}_{t-1}, \hat{X}_t\rangle > (1 - \frac{1}{\sqrt{\log T}})\Delta_t$. This implies that

$$\|\hat{X}_t\|_{V_{t-1}^{-1}} \geq \frac{(1 - \frac{1}{\sqrt{\log T}})\Delta_t}{\|\hat{\theta}_{t-1}\|_{V_{t-1}}} . \tag{2}$$

On the other hand, we claim that $\|\hat{\theta}_{t-1}\|_{V_{t-1}}$ can be upper bounded as $O(\sqrt{T})$. This can be seen from the fact that $\|\hat{\theta}_{t-1}\|_{V_{t-1}} = \|\hat{\theta}_{t-1} - \theta^* + \theta^*\|_{V_{t-1}} \leq \|\hat{\theta}_{t-1} - \theta^*\|_{V_{t-1}} + \|\theta^*\|_{V_{t-1}}$. Since the event $B_t$ holds, we know the first term is upper bounded by $\sqrt{\beta_{t-1}(\gamma)}$, and since the largest eigenvalue of the matrix $V_{t-1}$ is upper bounded by $\lambda + TL$ and $\|\theta^*\| \leq S$, the second term is upper bounded by $S\sqrt{\lambda + TL^2}$. Hence, $\|\hat{\theta}_{t-1}\|_{V_{t-1}}$ is upper bounded by $O(\sqrt{T})$. Then one can substitute this bound back into Eqn. (2), and this yields

$$\|\hat{X}_t\|_{V_{t-1}^{-1}} \geq \Omega\Big(\frac{1}{\sqrt{T}}\Big(1 - \frac{1}{\sqrt{\log T}}\Big)\Delta_t\Big) . \tag{3}$$

Furthermore, by our design of the algorithm, the index of $A_t$ is not larger than the index of the arm with the largest empirical reward at time $t$. Hence,

$$I_{t,A_t} = \frac{\hat{\Delta}_{t,A_t}^2}{\beta_{t-1}(\gamma)\|X_t\|_{V_{t-1}^{-1}}^2} + \log\frac{1}{\beta_{t-1}(\gamma)\|X_t\|_{V_{t-1}^{-1}}^2} \leq \log\frac{1}{\beta_{t-1}(\gamma)\|\hat{X}_t\|_{V_{t-1}^{-1}}^2} . \tag{4}$$

In the following, we set $\gamma$ as well as another free parameter $\Gamma$ as follows:

$$\Gamma = \frac{d\log^{\frac{3}{2}} T}{\sqrt{T}} \quad \text{and} \quad \gamma = \frac{1}{t^2} , . \tag{5}$$

If $\|X_t\|_{V_{t-1}^{-1}}^2 \geq \frac{\Delta_t^2}{\beta_{t-1}(\gamma)}$, by using Corollary 1 with the choice in Eqn. (5), the upper bound of $F_1$ in this case is $O(d\sqrt{T\log T})$. Otherwise, using the event $D_t$ and the bound in (3), we deduce that for all $T$ sufficiently large, we have $\|X_t\|_{V_{t-1}^{-1}}^2 \geq \Omega\big(\frac{\Delta_t^2}{\beta_{t-1}(\gamma)\log(T/\Delta_t^2)}\big)$. Therefore by using Corollary 1 and the "peeling device" (Lattimore & Szepesvári, 2020, Chapter 9) on $\Delta_t$ such that $2^{-l} < \Delta_t \leq 2^{-l+1}$ for $l = 1, 2, \ldots, \lceil Q\rceil$ where $Q = -\log_2 \Gamma$ and $\Gamma$ is chosen as in Eqn. (5). Now consider,

$$F_1 \leq O(1) + \mathbb{E}\sum_{t=1}^T \Delta_t \cdot \mathbb{1}\Big\{\|X_t\|_{V_{t-1}^{-1}}^2 \geq \Omega\Big(\frac{\Delta_t^2}{\beta_{t-1}(\gamma)\log(T/\Delta_t^2)}\Big)\Big\}$$

$$\leq O(1) + T\Gamma + \mathbb{E}\sum_{t=1}^T\sum_{l=1}^{\lceil Q\rceil} \Delta_t \cdot \mathbb{1}\Big\{\|X_t\|_{V_{t-1}^{-1}}^2 \geq \Omega\Big(\frac{\Delta_t^2}{\beta_{t-1}(\gamma)\log(T/\Delta_t^2)}\Big)\Big\} \mathbb{1}\{2^{-l} < \Delta_t \leq 2^{-l+1}\}$$

$$\leq O(1) + T\Gamma + \mathbb{E}\sum_{t=1}^T\sum_{l=1}^{\lceil Q\rceil} 2^{-l+1} \cdot \mathbb{1}\Big\{\|X_t\|_{V_{t-1}^{-1}}^2 \geq \Omega\Big(\frac{2^{-2l}}{\beta_{t-1}(\gamma)\log(T\cdot 2^{2l})}\Big)\Big\}$$

$$\leq O(1)+T\Gamma+\mathbb{E}\sum_{l=1}^{\lceil Q\rceil}2^{-l+1}O\left(2^{2l}d\beta_T(\gamma)\log(2^{2l}T)\log\left(1+\frac{2L^2\cdot 2^{2l}\beta_T(\gamma)\log(T\cdot 2^{2l})}{\lambda}\right)\right) \quad (6)$$

$$\leq O(1)+T\Gamma+\sum_{l=1}^{\lceil Q\rceil}2^{l+1}\cdot O\left(d\beta_T(\gamma)\log(\frac{T}{\Gamma^2})\log\left(1+\frac{L^2\beta_T(\gamma)\log(\frac{T}{\Gamma^2})}{\lambda\Gamma^2}\right)\right)$$

$$\leq O(1)+T\Gamma+O\left(\frac{d\beta_T(\gamma)\log(\frac{T}{\Gamma^2})}{\Gamma}\log\left(1+\frac{L^2\beta_T(\gamma)\log(\frac{T}{\Gamma^2})}{\lambda\Gamma^2}\right)\right), \quad (7)$$

where in Inequality (6) we used Corollary 1. Substituting the choices of $\Gamma$ and $\gamma$ in (5) into (7) yields the upper bound on $\mathbb{E}\sum_{t=1}^{T}\Delta_t\cdot\mathbb{1}\{B_t,C_t,D_t\}\cdot\mathbb{1}\{\|X_t\|^2_{V_{t-1}^{-1}}<\frac{\Delta_t^2}{\beta_{t-1}(\gamma)}\}$ of the order $O(d\sqrt{T}\log^{\frac{3}{2}}T)$. Hence $F_1\leq O(d\sqrt{T}\log^{\frac{3}{2}}T)$. Other details are fleshed out in Appendix A.2.

For $F_2$, since $C_t$ and $\overline{D}_t$ together imply that $\langle\theta^*,x_t^*\rangle-\delta<\varepsilon+\langle\hat{\theta}_{t-1},X_t\rangle$, then using the choices of $\delta$ and $\varepsilon$, we have $\langle\hat{\theta}_{t-1}-\theta^*,X_t\rangle>\frac{\Delta_t}{\sqrt{\log T}}$. Substituting this into the event $B_t$ and using the Cauchy–Schwarz inequality, we have

$$\|X_t\|^2_{V_{t-1}^{-1}}\geq\frac{\Delta_t^2}{\beta_{t-1}(\gamma)\log T}.$$

Again applying the "peeling device" on $\Delta_t$ and Corollary 1, we can upper bound $F_2$ as follows:

$$F_2\leq T\Gamma+O\left(\frac{d\beta_T(\gamma)\log T}{\Gamma}\right)\log\left(1+\frac{L^2\beta_T(\gamma)\log T}{\lambda\Gamma^2}\right). \quad (8)$$

Then with the choice of $\Gamma$ and $\gamma$ as stated in (5), the upper bound of the $F_2$ is also of order $O(d\sqrt{T}\log^{\frac{3}{2}}T)$. More details of the calculation leading to Eqn. (8) are in Appendix A.3.

For $F_3$, this is the case when the best arm at time $t$ does not perform sufficiently well so that the empirically largest reward at time $t$ is far from the highest expected reward. One observes that minimizing $F_3$ results in a tradeoff with respect to $F_1$. On the event $\overline{C}_t$, we can again apply the "peeling device" on $\langle\theta^*,x_t^*\rangle-\langle\hat{\theta}_{t-1},x_t^*\rangle$ such that $\frac{q+1}{2}\delta\leq\langle\theta^*,x_t^*\rangle-\langle\hat{\theta}_{t-1},x_t^*\rangle<\frac{q+2}{2}\delta$ where $q\in\mathbb{N}$. Then using the fact that $I_{t,A_t}\leq I_{t,a_t^*}$, we have

$$\log\frac{1}{\beta_{t-1}(\gamma)\|X_t\|^2_{V_{t-1}^{-1}}}<\frac{q^2\delta^2}{4\beta_{t-1}(\gamma)\|x_t^*\|^2_{V_{t-1}^{-1}}}+\log\frac{1}{\beta_{t-1}(\gamma)\|x_t^*\|^2_{V_{t-1}^{-1}}}. \quad (9)$$

On the other hand, using the event $B_t$ and the Cauchy–Schwarz inequality, it holds that

$$\|x_t^*\|_{V_{t-1}^{-1}}\geq\frac{(q+1)\delta}{2\sqrt{\beta_{t-1}(\gamma)}}. \quad (10)$$

If $\|X_t\|^2_{V_{t-1}^{-1}}\geq\frac{\Delta_t^2}{\beta_{t-1}(\gamma)}$, the regret in this case is bounded by $O(d\sqrt{T\log T})$. Otherwise, combining Eqn. (9) and Eqn. (10) implies that

$$\|X_t\|^2_{V_{t-1}^{-1}}\geq\frac{(q+1)^2\delta^2}{4\beta_{t-1}(\gamma)}\exp\left(-\frac{q^2}{(q+1)^2}\right).$$

Using Corollary 1, we can now conclude that $F_3$ is upper bounded as

$$F_3\leq T\Gamma+O\left(\frac{d\beta_T(\gamma)\log T}{\Gamma}\right)\log\left(1+\frac{L^2\beta_T(\gamma)\log T}{\lambda\Gamma^2}\right). \quad (11)$$

Substituting $\Gamma$ and $\gamma$ in Eqn. (5) into Eqn. (11), we can upper bound $F_3$ by $O(d\sqrt{T}\log^{\frac{3}{2}}T)$. Complete details are provided in Appendix A.4.

For $F_4$, using Lemma 1 with the choice of $\gamma=1/t^2$ and $Q=-\log\Gamma$, we have

$$F_4=\mathbb{E}\sum_{t=1}^{T}\Delta_t\cdot\mathbb{1}\{\overline{B}_t\}\leq T\Gamma+\mathbb{E}\sum_{t=1}^{T}\sum_{l=1}^{\lceil Q\rceil}\Delta_t\cdot\mathbb{1}\{2^{-l}<\Delta_t\leq 2^{-l+1}\}\mathbb{1}\{\overline{B}_t\}$$

$$\leq T\Gamma+\sum_{t=1}^{T}\sum_{l=1}^{\lceil Q\rceil}2^{-l+1}\mathbb{P}(\overline{B}_t)\leq T\Gamma+\sum_{t=1}^{T}\sum_{l=1}^{\lceil Q\rceil}2^{-l+1}\gamma<T\Gamma+\frac{\pi^2}{3}.$$

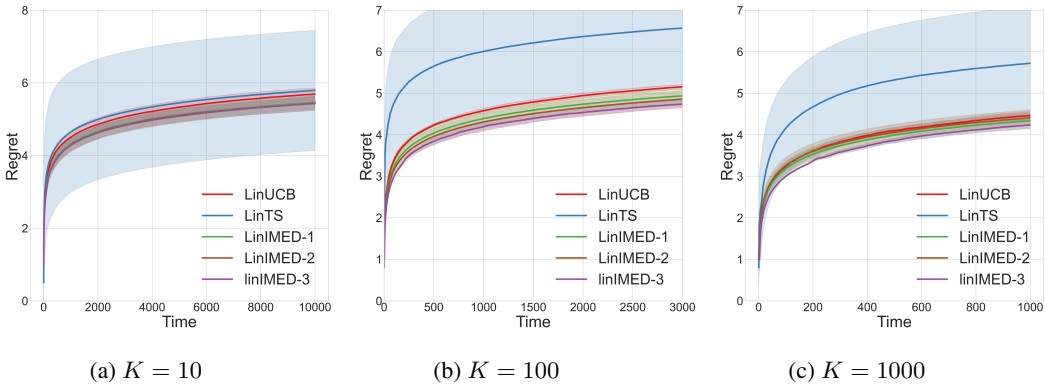

(a) $K = 10$        (b) $K = 100$        (c) $K = 1000$

Figure 1: Simulation results (expected regrets) on the synthetic dataset with different $K$'s

Thus, $F_4 \leq O(d\sqrt{T}\log^{\frac{3}{2}} T)$. In conclusion, with the choice of $\Gamma$ and $\gamma$ in Eqn. (5), we have shown that the expected regret of LinIMED-1 $R_T = \sum_{i=1}^{4} F_i$ is upper bounded by $O(d\sqrt{T}\log^{\frac{3}{2}} T)$.    □

**Remark 1.** *For LinIMED-2, the proof is similar but the assumption that $\langle \theta^*, x_{t,a}\rangle \geq 0$ is not required. For LinIMED-3, by directly using the UCB, we improve the regret bound to match the state-of-the-art $O(d\sqrt{T}\log T)$, which matches that of LinUCB with OFUL's confidence bound.*

## 6   EMPIRICAL STUDIES

This section aims to justify the utility of the family of LinIMED algorithms we developed and to demonstrate their effectiveness through quantitative evaluations in simulated environments and real-world datasets such as the MovieLens dataset.

We compare our LinIMED algorithms with LinTS and LinUCB with the choice $\lambda = L^2$. We set $\beta_t(\gamma) = (R\sqrt{3d\log(1+t)} + \sqrt{2})^2$ (here $\gamma = \frac{1}{(1+t)^2}$ and $L = \sqrt{2}$) for the synthetic dataset with varying and finite arm set and $\beta_t(\gamma) = (R\sqrt{d\log((1+t)t^2)} + \sqrt{20})^2$ (here $\gamma = \frac{1}{t^2}$ and $L = \sqrt{20}$) for the MovieLens dataset respectively. The confidence widths $\sqrt{\beta_t(\gamma)}$ for each algorithm are multiplied by a factor $\alpha$ and we tune $\alpha$ by searching over the grid $\{0.1, 0.2, \ldots, 1.0\}$ and report the **best performance** for each algorithm; see Appendix D. Both $\gamma$'s are of order $O(\frac{1}{t^2})$ as suggested by our proof sketch in Eqn. (5). We set $C = 30$ in LinIMED-3 throughout. The sub-Gaussian noise level is $R = 0.1$. We choose LinUCB and LinTS as competing algorithms since they are paradigmatic examples of deterministic and randomized contextual linear bandit algorithms respectively.

### 6.1   EXPERIMENTS ON A SYNTHETIC DATASET IN THE VARYING ARM SET SETTING

We perform an empirical study on a varying arm setting. We evaluate the performance with different dimensions $d$ and different number of arms $K$. We set the unknown parameter vector and the best context vector as $\theta^* = x_t^* = [\frac{1}{\sqrt{d-1}}, \ldots, \frac{1}{\sqrt{d-1}}, 0]^\top \in \mathbb{R}^d$. There are $K - 2$ suboptimal arms vectors, which are all the same (i.e., repeated) and share the context $[\frac{t}{(t+1)\sqrt{d-1}}, \ldots, \frac{t}{(t+1)\sqrt{d-1}}, \frac{t}{t+1}]^\top \in \mathbb{R}^d$. Finally, there is also one "worst" arm vector with context $[0, 0, \ldots, 0, 1]^\top$. This synthetic dataset is inspired by the synthetic varying arm set in Gales et al. (2022). First we fix $d = 20$. The results for different numbers of arms such as $K = 10, 100, 1000$ are shown in Fig. 1. Note that each plot is repeated 10 times to obtain the mean and standard deviation of the regret. From Fig. 1, we observe that LinIMED and its variants outperform LinTS and LinUCB regardless of the number of the arms $K$. Second, we set $K = 10$ with the dimension $d = 20, 50, 100$. Each trial is again repeated 10 times and the regret over time is shown in Fig. 2. Again, we see that LinIMED and its variants clearly perform better than LinUCB and LinTS.

The experimental results on synthetic data demonstrate that the performances of all three versions of the LinIMED algorithms are largely similar but LinIMED-3 is slightly superior (corroborating our theoretical findings). More importantly, they outperform both the LinTS and LinUCB algorithms in a statistically significant manner, regardless of the number of arms $K$ or the dimension $d$ of the data.

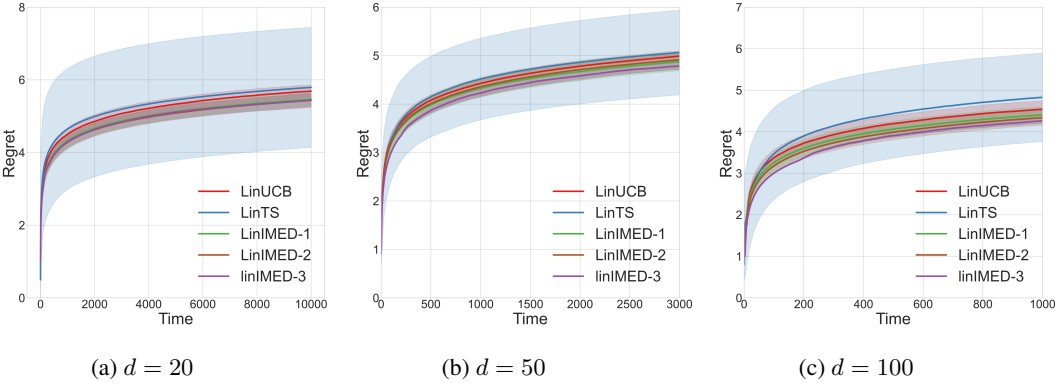

(a) $d = 20$    (b) $d = 50$    (c) $d = 100$

Figure 2: Simulation results (expected regrets) on the synthetic dataset with different $d$'s

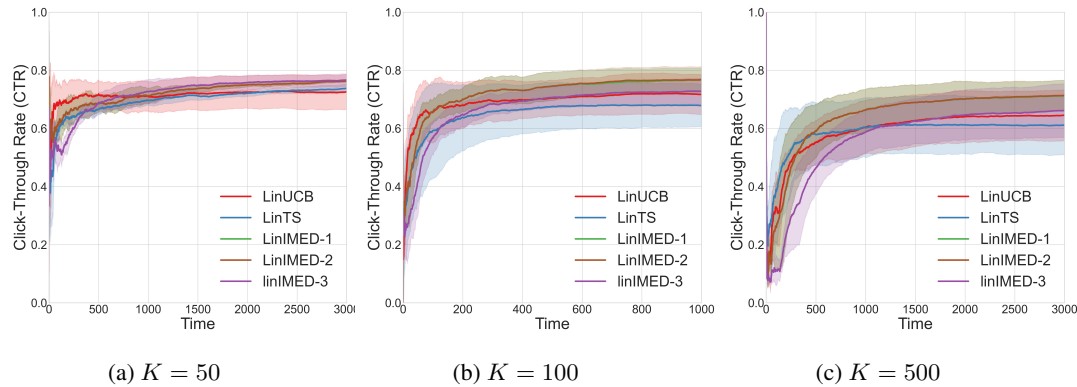

(a) $K = 50$    (b) $K = 100$    (c) $K = 500$

Figure 3: Simulation results (CTRs) of the MovieLens dataset with different $K$'s

## 6.2 Experiments on the MovieLens Dataset

The MovieLens dataset (Cantador et al. (2011)) is a widely-used benchmark dataset for research in recommendation systems. We specifically use the MovieLens 10M dataset, which contains 10 million ratings (from 0 to 5) and 100,000 tag applications applied to 10,000 movies by 72,000 users. To preprocess the dataset, we choose the best $K \in \{50, 100, 500\}$ movies for consideration. At each iteration $t$, one random user visits the website and is recommended one of the best $K$ movies. We assume that the user will click on the recommended movie if the user's rating of this movie is at least 3; otherwise, the user will not click. We implement the three versions of LinIMED, LinUCB, and LinTS on this dataset. Each trial is repeated over 5 runs and the averages and standard deviations of the click-through rates (CTRs) as functions of time are reported in Fig. 3. Note that the CTR is the number of clicks per time which corresponds to the reward. One observes that on the MovieLens dataset, the variants of LinIMED also perform similarly. Furthermore, they significantly outperform LinUCB and LinTS for all $K \in \{50, 100, 500\}$ when time horizon $T$ is sufficiently large. In particular, for $K \geq 100$, the CTR of LinIMED-1,2 is at least 0.05 larger than the traditional competitors.

## 7 Future Work

In the future, a fruitful direction of research is to analyze the effect of directly using the KL-divergence in the Line 14 of Algorithm 1 instead of the estimated squared gap $\hat{\Delta}_{t,a}^2$; we believe that in this case, the analysis would be more challenging, but the theoretical and empirical performances might be superior to our three LinIMED algorithms. In addition, one can generalize the family of IMED-style algorithms to generalized linear bandits or neural contextual bandits.

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
