# Supplementary Material for the ICLR 2024 submission "Linear Indexed Minimum Empirical Divergence Algorithms"

## A    PROOF OF THE REGRET BOUND FOR LINIMED-1 (COMPLETE PROOF OF THEOREM 1)

Here and in the following, we abbreviate $\beta_t(\gamma)$ as $\beta_t$, i.e., we drop the dependence of $\beta_t$ on $\gamma$, which is taken to be $\frac{1}{t^2}$ per Eqn. (5).

### A.1    STATEMENT OF LEMMAS FOR LINIMED-1

We first state the following lemmas which respectively show the upper bound of $F_1$ to $F_4$:

**Lemma 2.** *Under Assumption 1, the assumption that $\langle \theta^*, x_{t,a} \rangle \geq 0$ for all $t \geq 1$ and $a \in \mathcal{A}_t$, and the assumption that $\sqrt{\lambda} S \geq 1$, then for the free parameter $0 < \Gamma < 1$, the term $F_1$ for LinIMED-1 satisfies:*

$$F_1 \leq O(1) + T\Gamma + O\left( \frac{d\beta_T \log(\frac{T}{\Gamma^2})}{\Gamma} \log\left( 1 + \frac{L^2 \beta_T \log(\frac{T}{\Gamma^2})}{\lambda \Gamma^2} \right) \right). \tag{12}$$

*With the choice of $\Gamma$ as in Eqn. (5),*

$$F_1 \leq O\left( d\sqrt{T} \log^{\frac{3}{2}} T \right).$$

**Lemma 3.** *Under Assumption 1, and the assumption that $\sqrt{\lambda} S \geq 1$, for the free parameter $0 < \Gamma < 1$, the term $F_2$ for LinIMED-1 satisfies:*

$$F_2 \leq 2T\Gamma + O\left( \frac{d\beta_T \log T}{\Gamma} \right) \log\left( 1 + \frac{L^2 \beta_T \log T}{\lambda \Gamma^2} \right). \tag{13}$$

*With the choice of $\Gamma$ as in Eqn. (5),*

$$F_2 \leq O\left( d\sqrt{T} \log^{\frac{3}{2}} T \right).$$

**Lemma 4.** *Under Assumption 1, and the assumption that $\sqrt{\lambda} S \geq 1$, for the free parameter $0 < \Gamma < 1$, the term $F_3$ for LinIMED-1 satisfies:*

$$F_3 \leq 2T\Gamma + O\left( \frac{d\beta_T \log(T)}{\Gamma} \log\left( 1 + \frac{L^2 \beta_T \log(T)}{\lambda \Gamma^2} \right) \right). \tag{14}$$

*With the choice of $\Gamma$ as in Eqn. (5),*

$$F_3 \leq O\left( d\sqrt{T} \log^{\frac{3}{2}} T \right).$$

**Lemma 5.** *Under Assumption 1, for the free parameter $0 < \Gamma < 1$, the term $F_4$ for LinIMED-1 satisfies:*

$$F_4 \leq T\Gamma + O(1).$$

*With the choice of $\Gamma$ as in Eqn. (5),*

$$F_4 \leq O\left( d\sqrt{T} \log^{\frac{3}{2}} T \right).$$

### A.2    PROOF OF LEMMA 2

*Proof.* From the event $C_t$ and the fact that $\langle \theta^*, x_t^* \rangle = \Delta_t + \langle \theta^*, X_t \rangle \geq \Delta_t$ (here is where we use that $\langle \theta^*, x_{t,a} \rangle \geq 0$ for all $t$ and $a$), we obtain $\max_{b \in \mathcal{A}_t} \langle \hat{\theta}_{t-1}, x_{t,b} \rangle > (1 - \frac{1}{\sqrt{\log T}}) \Delta_t$. For convenience, define $\hat{A}_t := \arg\max_{b \in \mathcal{A}_t} \langle \hat{\theta}_{t-1}, x_{t,b} \rangle$ as the empirically best arm at time step $t$, where ties are

broken arbitrarily, then use $\hat{X}_t$ to denote the corresponding context of the arm $\hat{A}_t$. Therefore from the Cauchy–Schwarz inequality, we have $\|\hat{\theta}_{t-1}\|_{V_{t-1}}\|\hat{X}_t\|_{V_{t-1}^{-1}} \geq \langle\hat{\theta}_{t-1}, \hat{X}_t\rangle > (1 - \frac{1}{\sqrt{\log T}})\Delta_t$. This implies that

$$\|\hat{X}_t\|_{V_{t-1}^{-1}} \geq \frac{(1 - \frac{1}{\sqrt{\log T}})\Delta_t}{\|\hat{\theta}_{t-1}\|_{V_{t-1}}} .$$

On the other hand, we claim that $\|\hat{\theta}_{t-1}\|_{V_{t-1}}$ can be upper bounded as $O(\sqrt{T})$. This can be seen from the fact that $\|\hat{\theta}_{t-1}\|_{V_{t-1}} = \|\hat{\theta}_{t-1} - \theta^* + \theta^*\|_{V_{t-1}} \leq \|\hat{\theta}_{t-1} - \theta^*\|_{V_{t-1}} + \|\theta^*\|_{V_{t-1}}$. Since the event $B_t$ holds, we know the first term is upper bounded by $\sqrt{\beta_{t-1}(\gamma)}$, and since the maximum eigenvalue of the matrix $V_{t-1}$ is upper bounded by $\lambda + TL$ and $\|\theta^*\| \leq S$, the second term is upper bounded by $S\sqrt{\lambda + TL}$. Hence, $\|\hat{\theta}_{t-1}\|_{V_{t-1}}$ is upper bounded by $O(\sqrt{T})$. Then one can substitute this bound back into Eqn. (2), and this yields

$$\|\hat{X}_t\|_{V_{t-1}^{-1}} \geq \Omega\left(\frac{1}{\sqrt{T}}\left(1 - \frac{1}{\sqrt{\log T}}\right)\Delta_t\right) .$$

Furthermore, by our design of the algorithm, the index of $A_t$ is not larger than the index of the arm with the largest empirical reward at time $t$. Hence,

$$I_{t,A_t} = \frac{\hat{\Delta}_{t,A_t}^2}{\beta_{t-1}(\gamma)\|X_t\|_{V_{t-1}^{-1}}^2} + \log\frac{1}{\beta_{t-1}(\gamma)\|X_t\|_{V_{t-1}^{-1}}^2} \leq \log\frac{1}{\beta_{t-1}(\gamma)\|\hat{X}_t\|_{V_{t-1}^{-1}}^2} .$$

If $\|X_t\|_{V_{t-1}^{-1}}^2 \geq \frac{\Delta_t^2}{\beta_{t-1}}$, by using Corollary 1 with the choice of parameters as in Eqn. (5),

$$\mathbb{E}\sum_{t=1}^{T}\Delta_t \cdot \mathbb{1}\{B_t, C_t, D_t\} \cdot \mathbb{1}\left\{\|X_t\|_{V_{t-1}^{-1}}^2 \geq \frac{\Delta_t^2}{\beta_{t-1}}\right\} \leq \mathbb{E}\sum_{t=1}^{T}\Delta_t \cdot \mathbb{1}\left\{\|X_t\|_{V_{t-1}^{-1}}^2 \geq \frac{\Delta_t^2}{\beta_{t-1}}\right\} \quad (15)$$

$$\leq T\Gamma + \mathbb{E}\sum_{t=1}^{T}\sum_{l=1}^{\lceil Q\rceil}\Delta_t \cdot \mathbb{1}\left\{\|X_t\|_{V_{t-1}^{-1}}^2 \geq \frac{\Delta_t^2}{\beta_{t-1}}\right\} \cdot \mathbb{1}\left\{2^{-l} < \Delta_t \leq 2^{-l+1}\right\}$$

$$\leq T\Gamma + \mathbb{E}\sum_{t=1}^{T}\sum_{l=1}^{\lceil Q\rceil}2^{-l+1} \cdot \mathbb{1}\left\{\|X_t\|_{V_{t-1}^{-1}}^2 \geq \frac{2^{-2l}}{\beta_T}\right\}$$

$$\leq T\Gamma + \mathbb{E}\sum_{l=1}^{\lceil Q\rceil}2^{-l+1}\frac{6d\beta_T}{2^{-2l}}\log\left(1 + \frac{2L^2\beta_T}{\lambda \cdot 2^{-2l}}\right)$$

$$= T\Gamma + \mathbb{E}\sum_{l=1}^{\lceil Q\rceil}2^l \cdot 12d\beta_T\log\left(1 + \frac{2^{2l+1}L^2\beta_T}{\lambda}\right)$$

$$< T\Gamma + \mathbb{E}\sum_{l=1}^{\lceil Q\rceil}2^l \cdot 12d\beta_T\log\left(1 + \frac{2^{2Q+3}L^2\beta_T}{\lambda}\right)$$

$$= T\Gamma + (2^{\lceil Q\rceil} - 1) \cdot 24d\beta_T\log\left(1 + \frac{2^{2Q+3}L^2\beta_T}{\lambda}\right)$$

$$< T\Gamma + \frac{48d\beta_T}{\Gamma}\log\left(1 + \frac{8L^2\beta_T}{\lambda\Gamma^2}\right)$$

Then with the choice of $\Gamma$ as in Eqn. (5),

$$\mathbb{E}\sum_{t=1}^{T}\Delta_t \cdot \mathbb{1}\{B_t, C_t, D_t\} \cdot \mathbb{1}\left\{\|X_t\|_{V_{t-1}^{-1}}^2 \geq \frac{\Delta_t^2}{\beta_{t-1}}\right\}$$

$$< d\sqrt{T}\log^{\frac{3}{2}}T + \frac{48\beta_T\sqrt{T}}{\log^{\frac{3}{2}}T}\log\left(1 + \frac{8L^2\beta_T T}{\lambda d^2\log^3 T}\right)$$

$$\leq O\left(d\sqrt{T\log T}\right) . \quad (16)$$

Otherwise we have $\|X_t\|^2_{V^{-1}_{t-1}} < \frac{\Delta^2_t}{\beta_{t-1}}$, then $\log \frac{1}{\beta_{t-1}\|X_t\|^2_{V^{-1}_{t-1}}} > 0$ since $\Delta_t \leq 1$. Substituting this into Eqn. (4), then using the event $D_t$ and the bound in (3), we deduce that for all $T$ sufficiently large, we have $\|X_t\|^2_{V^{-1}_{t-1}} \geq \Omega\big(\frac{\Delta^2_t}{\beta_{t-1}\log(T/\Delta^2_t)}\big)$. Therefore by using Corollary 1 and the "peeling device" (Lattimore & Szepesvári, 2020, Chapter 9) on $\Delta_t$ such that $2^{-l} < \Delta_t \leq 2^{-l+1}$ for $l = 1, 2, \ldots, \lceil Q \rceil$ where $\Gamma := 2^{-Q}$ is a free parameter that we can choose. Consider,

$$\mathbb{E}\sum_{t=1}^{T}\Delta_t \cdot \mathbb{1}\{B_t, C_t, D_t\} \cdot \mathbb{1}\left\{\|X_t\|^2_{V^{-1}_{t-1}} < \frac{\Delta^2_t}{\beta_{t-1}}\right\}$$

$$\leq \mathbb{E}\sum_{t=1}^{T}\Delta_t \cdot \mathbb{1}\{B_t, C_t, D_t\} \cdot \mathbb{1}\left\{\Delta_t \leq 2^{-\lceil Q \rceil}\right\} \cdot \mathbb{1}\left\{\|X_t\|^2_{V^{-1}_{t-1}} < \frac{\Delta^2_t}{\beta_{t-1}}\right\}$$

$$+ \mathbb{E}\sum_{t=1}^{T}\Delta_t \cdot \mathbb{1}\{B_t, C_t, D_t\} \cdot \mathbb{1}\left\{\Delta_t > 2^{-\lceil Q \rceil}\right\} \cdot \mathbb{1}\left\{\|X_t\|^2_{V^{-1}_{t-1}} < \frac{\Delta^2_t}{\beta_{t-1}}\right\}$$

$$\leq O(1) + T\Gamma + \mathbb{E}\sum_{t=1}^{T}\Delta_t \cdot \mathbb{1}\left\{\|X_t\|^2_{V^{-1}_{t-1}} \geq \Omega\big(\frac{\Delta^2_t}{\beta_{t-1}\log(T/\Delta^2_t)}\big)\right\}\mathbb{1}\left\{\Delta_t > 2^{-\lceil Q \rceil}\right\}$$

$$\leq O(1) + T\Gamma + \mathbb{E}\sum_{t=1}^{T}\sum_{l=1}^{\lceil Q \rceil}\Delta_t \cdot \mathbb{1}\left\{\|X_t\|^2_{V^{-1}_{t-1}} \geq \Omega\big(\frac{\Delta^2_t}{\beta_{t-1}\log(T/\Delta^2_t)}\big)\right\}\mathbb{1}\{2^{-l} < \Delta_t \leq 2^{-l+1}\}$$

$$\leq O(1) + T\Gamma + \mathbb{E}\sum_{t=1}^{T}\sum_{l=1}^{\lceil Q \rceil}2^{-l+1} \cdot \mathbb{1}\left\{\|X_t\|^2_{V^{-1}_{t-1}} \geq \Omega\big(\frac{2^{-2l}}{\beta_{t-1}\log(T \cdot 2^{2l})}\big)\right\}$$

$$= O(1) + T\Gamma + \mathbb{E}\sum_{l=1}^{\lceil Q \rceil}2^{-l+1}\sum_{t=1}^{T} \mathbb{1}\left\{\|X_t\|^2_{V^{-1}_{t-1}} \geq \Omega\big(\frac{2^{-2l}}{\beta_{t-1}\log(T \cdot 2^{2l})}\big)\right\}$$

$$\leq O(1) + T\Gamma + \mathbb{E}\sum_{l=1}^{\lceil Q \rceil}2^{-l+1}O\Big(2^{2l}d\beta_T\log(T \cdot 2^{2l})\log\Big(1 + \frac{2L^2 \cdot 2^{2l}\beta_T\log(T \cdot 2^{2l})}{\lambda}\Big)\Big)$$

$$< O(1) + T\Gamma + \mathbb{E}\sum_{l=1}^{\lceil Q \rceil}2^{l+1} \cdot O\Big(d\beta_T\log(\frac{T}{\Gamma^2})\log\Big(1 + \frac{L^2\beta_T\log(\frac{T}{\Gamma^2})}{\lambda\Gamma^2}\Big)\Big)$$

$$\leq O(1) + T\Gamma + O\Big(\frac{d\beta_T\log(\frac{T}{\Gamma^2})}{\Gamma}\log\Big(1 + \frac{L^2\beta_T\log(\frac{T}{\Gamma^2})}{\lambda\Gamma^2}\Big)\Big),$$

This proves Eqn. (12). Then with the choice of the parameters as in Eqn. (5),

$$\mathbb{E}\sum_{t=1}^{T}\Delta_t \cdot \mathbb{1}\{B_t, C_t, D_t\} \cdot \mathbb{1}\left\{\|X_t\|^2_{V^{-1}_{t-1}} < \frac{\Delta^2_t}{\beta_{t-1}}\right\}$$

$$< O(1) + d\sqrt{T}\log^{\frac{3}{2}}T + O\Big(d\beta_T\log\Big(\frac{T^2}{d^2\log^3 T}\Big)\frac{\sqrt{T}}{d\log^{\frac{3}{2}}T}\log\Big(1 + \frac{L^2\beta_T T}{\lambda d^2\log^3 T} \cdot \log\Big(\frac{T^2}{d^2\log^3 T}\Big)\Big)\Big)$$

$$\leq O\Big(d\sqrt{T}\log^{\frac{3}{2}}T\Big).$$

Hence, we can upper bound $F_1$ as

$$F_1 = \mathbb{E}\sum_{t=1}^{T}\Delta_t \cdot \mathbb{1}\{B_t, C_t, D_t\} \cdot \mathbb{1}\left\{\|X_t\|^2_{V^{-1}_{t-1}} \geq \frac{\Delta^2_t}{\beta_{t-1}}\right\} + \mathbb{E}\sum_{t=1}^{T}\Delta_t \cdot \mathbb{1}\{B_t, C_t, D_t\} \cdot \mathbb{1}\left\{\|X_t\|^2_{V^{-1}_{t-1}} < \frac{\Delta^2_t}{\beta_{t-1}}\right\}$$

$$\leq O\Big(d\sqrt{T\log T}\Big) + O\Big(d\sqrt{T}\log^{\frac{3}{2}}T\Big)$$

$$\leq O\Big(d\sqrt{T}\log^{\frac{3}{2}}T\Big),$$

which concludes the proof. □

### A.3 PROOF OF LEMMA 3

*Proof.* Since $C_t$ and $\overline{D}_t$ together imply that $\langle \theta^*, x_t^* \rangle - \delta < \varepsilon + \langle \hat{\theta}_{t-1}, X_t \rangle$, then using the choices of $\delta$ and $\varepsilon$, we have $\langle \hat{\theta}_{t-1} - \theta^*, X_t \rangle > \frac{\Delta_t}{\sqrt{\log T}}$. Substituting this into the event $B_t$ and using the Cauchy–Schwarz inequality, we have

$$\|X_t\|_{V_{t-1}^{-1}}^2 \geq \frac{\Delta_t^2}{\beta_{t-1}(\gamma) \log T}.$$

Again applying the "peeling device" on $\Delta_t$ and Corollary 1, we can upper bound $F_2$ as follows:

$$F_2 \leq \mathbb{E} \sum_{t=1}^{T} \Delta_t \cdot \mathbb{1} \left\{ \|X_t\|_{V_{t-1}^{-1}}^2 \geq \frac{\Delta_t^2}{\beta_{t-1} \log T} \right\}$$

$$\leq T\Gamma + \mathbb{E} \sum_{t=1}^{T} \sum_{l=1}^{\lceil Q \rceil} \Delta_t \cdot \mathbb{1} \left\{ \|X_t\|_{V_{t-1}^{-1}}^2 \geq \frac{\Delta_t^2}{\beta_{t-1} \log T} \right\} \cdot \mathbb{1} \left\{ 2^{-l} < \Delta_t \leq 2^{-l+1} \right\}$$

$$\leq T\Gamma + \mathbb{E} \sum_{t=1}^{T} \sum_{l=1}^{\lceil Q \rceil} 2^{-l+1} \cdot \mathbb{1} \left\{ \|X_t\|_{V_{t-1}^{-1}}^2 \geq \frac{2^{-2l}}{\beta_T \log T} \right\}$$

$$\leq T\Gamma + \mathbb{E} \sum_{l=1}^{\lceil Q \rceil} 2^{-l+1} \cdot 2^{2l} \cdot 6d\beta_T (\log T) \log \left( 1 + \frac{2^{2l+1} \cdot L^2 \beta_T \log T}{\lambda} \right)$$

$$\leq T\Gamma + \mathbb{E} \sum_{l=1}^{\lceil Q \rceil} 2^{l} \cdot 12d\beta_T (\log T) \log \left( 1 + \frac{2^{2\lceil Q \rceil+1} \cdot L^2 \beta_T \log T}{\lambda} \right)$$

$$= T\Gamma + (2^{\lceil Q \rceil} - 1) \cdot 24d\beta_T (\log T) \log \left( 1 + \frac{2^{2\lceil Q \rceil+1} \cdot L^2 \beta_T \log T}{\lambda} \right)$$

$$< T\Gamma + \frac{48d\beta_T \log T}{\Gamma} \log \left( 1 + \frac{8L^2 \beta_T \log T}{\lambda \Gamma^2} \right)$$

$$= T\Gamma + O\left( \frac{d\beta_T \log T}{\Gamma} \log \left( 1 + \frac{L^2 \beta_T \log T}{\lambda \Gamma^2} \right) \right)$$

This proves Eqn. (13). Hence with the choice of the parameter $\Gamma$ as in Eqn. (5),

$$F_2 \leq d\sqrt{T} \log^{\frac{3}{2}} T + O\left( d\sqrt{T} \log^{\frac{3}{2}} T \right)$$

$$\leq O\left( d\sqrt{T} \log^{\frac{3}{2}} T \right).$$

□

### A.4 PROOF OF LEMMA 4

*Proof.* For $F_3$, this is the case when the best arm at time $t$ does not perform sufficiently well so that the empirically largest reward at time $t$ is far from the highest expected reward. One observes that minimizing $F_3$ results in a tradeoff with respect to $F_1$. On the event $\overline{C}_t$, we can apply the "peeling device" on $\langle \theta^*, x_t^* \rangle - \langle \hat{\theta}_{t-1}, x_t^* \rangle$ such that $\frac{q+1}{2}\delta \leq \langle \theta^*, x_t^* \rangle - \langle \hat{\theta}_{t-1}, x_t^* \rangle < \frac{q+2}{2}\delta$ where $q \in \mathbb{N}$. Then using the fact that $I_{t,A_t} \leq I_{t,a_t^*}$, we have

$$\log \frac{1}{\beta_{t-1} \|X_t\|_{V_{t-1}^{-1}}^2} < \frac{q^2 \delta^2}{4\beta_{t-1} \|x_t^*\|_{V_{t-1}^{-1}}^2} + \log \frac{1}{\beta_{t-1} \|x_t^*\|_{V_{t-1}^{-1}}^2}. \quad (17)$$

On the other hand, using the event $B_t$ and the Cauchy–Schwarz inequality, it holds that

$$\|x_t^*\|_{V_{t-1}^{-1}} \geq \frac{(q+1)\delta}{2\sqrt{\beta_{t-1}}}. \quad (18)$$

If $\|X_t\|^2_{V_{t-1}^{-1}} \geq \frac{\Delta_t^2}{\beta_{t-1}}$, the regret in this case is bounded by $O(d\sqrt{T \log T})$ (similar to the procedure to get from Eqn. (15) to Eqn. (16)). Otherwise $\log \frac{1}{\beta_{t-1}\|X_t\|^2_{V_{t-1}^{-1}}} > \log \frac{1}{\Delta_t^2} \geq 0$, then combining Eqn. (17) and Eqn. (18) implies that

$$\|X_t\|^2_{V_{t-1}^{-1}} \geq \frac{(q+1)^2 \delta^2}{4\beta_{t-1}} \exp\left(-\frac{q^2}{(q+1)^2}\right).$$

Notice here with $\sqrt{\lambda} S \geq 1$, $\|X_t\|^2_{V_{t-1}^{-1}} < \frac{\Delta_t^2}{\beta_{t-1}} \leq \frac{1}{\beta_{t-1}} \leq 1$, it holds that for all $q \in \mathbb{N}$,

$$\frac{(q+1)^2 \delta^2}{4\beta_{t-1}} \exp\left(-\frac{q^2}{(q+1)^2}\right) < 1. \tag{19}$$

Using Corollary 1, one can show that :

$$\sum_{t=1}^T \Delta_t \cdot \mathbb{1}\{B_t, \overline{C}_t\} \cdot \mathbb{1}\left\{\|X_t\|^2_{V_{t-1}^{-1}} < \frac{\Delta_t^2}{\beta_{t-1}}\right\}$$

$$\leq T\Gamma + \sum_{t=1}^T \sum_{l=1}^{\lceil Q \rceil} \Delta_t \cdot \mathbb{1}\{B_t, \overline{C}_t\} \cdot \mathbb{1}\left\{\|X_t\|^2_{V_{t-1}^{-1}} < \frac{\Delta_t^2}{\beta_{t-1}}\right\} \cdot \mathbb{1}\left\{2^{-l} < \Delta_t \leq 2^{-l+1}\right\}$$

$$\leq T\Gamma + \sum_{t=1}^T \sum_{l=1}^{\lceil Q \rceil} \sum_{q=1}^{\infty} \Delta_t \cdot \mathbb{1}\{B_t\} \cdot \mathbb{1}\left\{\frac{q+1}{2}\delta \leq \langle \theta^*, x_t^* \rangle - \langle \hat{\theta}_{t-1}, x_t^* \rangle < \frac{q+2}{2}\delta\right\} \cdot \mathbb{1}\left\{\|X_t\|^2_{V_{t-1}^{-1}} < \frac{\Delta_t^2}{\beta_{t-1}}\right\}$$

$$\cdot \mathbb{1}\left\{2^{-l} < \Delta_t \leq 2^{-l+1}\right\}$$

$$\leq T\Gamma + \sum_{t=1}^T \sum_{l=1}^{\lceil Q \rceil} \sum_{q=1}^{\infty} \Delta_t \cdot \mathbb{1}\left\{1 \geq \|X_t\|^2_{V_{t-1}^{-1}} \geq \frac{(q+1)^2 \delta^2}{4\beta_{t-1}} \exp\left(-\frac{q^2}{(q+1)^2}\right)\right\} \cdot \mathbb{1}\left\{2^{-l} < \Delta_t \leq 2^{-l+1}\right\}$$

$$= T\Gamma + \sum_{t=1}^T \sum_{l=1}^{\lceil Q \rceil} \sum_{q=1}^{\infty} \Delta_t \cdot \mathbb{1}\left\{1 \geq \|X_t\|^2_{V_{t-1}^{-1}} \geq \frac{(q+1)^2 \Delta_t^2}{4\beta_{t-1} \log T} \exp\left(-\frac{q^2}{(q+1)^2}\right)\right\} \cdot \mathbb{1}\left\{2^{-l} < \Delta_t \leq 2^{-l+1}\right\}$$

$$\leq T\Gamma + \sum_{t=1}^T \sum_{l=1}^{\lceil Q \rceil} \sum_{q=1}^{\infty} 2^{-l+1} \cdot \mathbb{1}\left\{1 \geq \|X_t\|^2_{V_{t-1}^{-1}} > \frac{(q+1)^2 \cdot 2^{-2l}}{4\beta_T \log T} \exp\left(-\frac{q^2}{(q+1)^2}\right)\right\}$$

$$\leq T\Gamma + \sum_{l=1}^{\lceil Q \rceil} \sum_{q=1}^{\infty} 2^{-l+1} \cdot 2^{2l} \cdot 24 d\beta_T (\log T) \cdot \frac{\exp\left(\frac{q^2}{(q+1)^2}\right)}{(q+1)^2} \cdot \log\left(1 + \frac{2^{2l} \cdot 8L^2 \beta_T \log T}{\lambda} \cdot \frac{\exp\left(\frac{q^2}{(q+1)^2}\right)}{(q+1)^2}\right)$$

$$< T\Gamma + \sum_{l=1}^{\lceil Q \rceil} \sum_{q=1}^{\infty} 2^{l+1} \cdot 24 d\beta_T (\log T) \cdot \frac{\exp\left(\frac{q^2}{(q+1)^2}\right)}{(q+1)^2} \cdot \log\left(1 + \frac{2^{2l+1} \cdot L^2 \beta_T \log T}{\lambda}\right)$$

$$= T\Gamma + \sum_{l=1}^{\lceil Q \rceil} 2^{l+1} \cdot 24 d\beta_T (\log T) \cdot \log\left(1 + \frac{2^{2l+1} \cdot L^2 \beta_T \log T}{\lambda}\right) \sum_{q=1}^{\infty} \frac{\exp\left(\frac{q^2}{(q+1)^2}\right)}{(q+1)^2}$$

$$\leq T\Gamma + \sum_{l=1}^{\lceil Q \rceil} 2^{l+1} \cdot 24 d\beta_T (\log T) \cdot \log\left(1 + \frac{2^{2l+1} \cdot L^2 \beta_T \log T}{\lambda}\right) \cdot (1.09)$$

$$\leq T\Gamma + \sum_{l=1}^{\lceil Q \rceil} 2^{l+1} \cdot 27 d\beta_T (\log T) \cdot \log\left(1 + \frac{2^{2l+1} \cdot L^2 \beta_T \log T}{\lambda}\right)$$

$$\leq T\Gamma + \sum_{l=1}^{\lceil Q \rceil} 2^{l+1} \cdot 27 d\beta_T (\log T) \cdot \log\left(1 + \frac{2^{2\lceil Q \rceil + 1} \cdot L^2 \beta_T \log T}{\lambda}\right)$$

$$< T\Gamma + \sum_{l=1}^{\lceil Q \rceil} \frac{216 d\beta_T \log T}{\Gamma} \cdot \log\left(1 + \frac{8L^2 \beta_T \log T}{\lambda \Gamma^2}\right)$$

$$= T\Gamma + O\left(\frac{d\beta_T \log T}{\Gamma} \log\left(1 + \frac{L^2 \beta_T \log T}{\lambda \Gamma^2}\right)\right). \tag{20}$$

Hence

$$F_3 = \sum_{t=1}^{T} \Delta_t \cdot \mathbb{1}\left\{B_t, \overline{C}_t\right\} \cdot \mathbb{1}\left\{\|X_t\|^2_{V_{t-1}^{-1}} < \frac{\Delta_t^2}{\beta_{t-1}}\right\} + \sum_{t=1}^{T} \Delta_t \cdot \mathbb{1}\left\{B_t, \overline{C}_t\right\} \cdot \mathbb{1}\left\{\|X_t\|^2_{V_{t-1}^{-1}} \geq \frac{\Delta_t^2}{\beta_{t-1}}\right\}$$

$$< O\left(\frac{d\beta_T}{\Gamma} \log\left(1 + \frac{L^2 \beta_T}{\lambda \Gamma^2}\right)\right) + 2T\Gamma + O\left(\frac{d\beta_T \log T}{\Gamma} \log\left(1 + \frac{L^2 \beta_T \log T}{\lambda \Gamma^2}\right)\right)$$

$$\leq 2T\Gamma + O\left(\frac{d\beta_T \log(T)}{\Gamma} \log\left(1 + \frac{L^2 \beta_T \log(T)}{\lambda \Gamma^2}\right)\right).$$

This proves Eqn. (14). With the choice of $\Gamma$ as in Eqn. (5),

$$F_3 \leq 2d\sqrt{T} \log^{\frac{3}{2}} T + O\left(\frac{d\sqrt{T}\beta_T \log T}{d \log^{\frac{3}{2}} T} \log\left(1 + \frac{TL^2 \beta_T \log T}{\lambda d^2 \log^3 T}\right)\right)$$

$$< 2d\sqrt{T} \log^{\frac{3}{2}} T + O\left(d\sqrt{T} \log^{\frac{3}{2}} T\right)$$

$$= O\left(d\sqrt{T} \log^{\frac{3}{2}} T\right).$$

$\square$

## A.5 PROOF OF LEMMA 5

*Proof.* For $F_4$, the proof is straightforward by using Lemma 1 with the choice of $\gamma$. Indeed, one has

$$F_4 = \mathbb{E}\sum_{t=1}^{T} \Delta_t \cdot \mathbb{1}\left\{\overline{B}_t\right\} \leq T\Gamma + \mathbb{E}\sum_{t=1}^{T}\sum_{l=1}^{\lceil Q \rceil} \Delta_t \cdot \mathbb{1}\left\{2^{-l} < \Delta_t \leq 2^{-l+1}\right\} \mathbb{1}\left\{\overline{B}_t\right\}$$

$$\leq T\Gamma + \mathbb{E}\sum_{t=1}^{T}\sum_{l=1}^{\lceil Q \rceil} 2^{-l+1} \mathbb{1}\left\{\overline{B}_t\right\} \leq T\Gamma + \sum_{t=1}^{T}\sum_{l=1}^{\lceil Q \rceil} 2^{-l+1} \mathbb{P}(\overline{B}_t) \leq T\Gamma + \sum_{t=1}^{T}\sum_{l=1}^{\lceil Q \rceil} 2^{-l+1}\gamma$$

$$= T\Gamma + \sum_{t=1}^{T} \frac{1}{t^2} \sum_{l=1}^{\lceil Q \rceil} 2^{-l+1} = T\Gamma + \sum_{t=1}^{T} \frac{2 - \Gamma}{t^2} < T\Gamma + \frac{\pi^2}{3} = T\Gamma + O(1).$$

With the choice of $\Gamma$ as in Eqn. (5),

$$F_4 < d\sqrt{T} \log^{\frac{3}{2}} T + O(1)$$

$$\leq O\left(d\sqrt{T} \log^{\frac{3}{2}} T\right).$$

$\square$

## A.6 PROOF OF THEOREM 1

*Proof.* Combining Lemmas 2, 3, 4 and 5,

$$R_T = F_1 + F_2 + F_3 + F_4$$

$$\leq O\left(d\sqrt{T} \log^{\frac{3}{2}} T\right) + O\left(d\sqrt{T} \log^{\frac{3}{2}} T\right) + O\left(d\sqrt{T} \log^{\frac{3}{2}} T\right) + O\left(d\sqrt{T} \log^{\frac{3}{2}} T\right)$$

$$= O\left(d\sqrt{T} \log^{\frac{3}{2}} T\right).$$

$\square$

# B  PROOF OF THE REGRET BOUND FOR LINIMED-2 (PROOF OF THEOREM 2)

We choose $\gamma$ and $\Gamma$ as follows:

$$\gamma = \frac{1}{t^2} \qquad \Gamma = \frac{\sqrt{d\beta_T}\log T}{\sqrt{T}} . \tag{21}$$

## B.1  STATEMENT OF LEMMAS FOR LINIMED-2

We first state the following lemmas which respectively show the upper bound of $F_1$ to $F_4$:

**Lemma 6.** *Under Assumption 1, and the assumption that $\sqrt{\lambda}S \geq 1$, for the free parameter $0 < \Gamma < 1$, the term $F_1$ for LinIMED-3 satisfies:*

$$F_1 \leq T\Gamma + O\left(\frac{d\beta_T \log T}{\Gamma}\right)\log\left(1 + \frac{L^2\beta_T \log T}{\lambda\Gamma^2}\right) .$$

**Lemma 7.** *Under Assumption 1, and the assumption that $\sqrt{\lambda}S \geq 1$, for the free parameter $0 < \Gamma < 1$, the term $F_2$ for LinIMED-3 satisfies:*

$$F_2 \leq T\Gamma + O\left(\frac{d\beta_T \log T}{\Gamma}\right)\log\left(1 + \frac{L^2\beta_T \log T}{\lambda\Gamma^2}\right) .$$

**Lemma 8.** *Under Assumption 1, and the assumption that $\sqrt{\lambda}S \geq 1$, for the free parameter $0 < \Gamma < 1$, the term $F_3$ for LinIMED-3 satisfies:*

$$F_3 \leq 5T\Gamma + O\left(\frac{d\beta_T \log T}{\Gamma}\log\left(1 + \frac{L^2\beta_T \log T}{\lambda\Gamma^2}\right)\right) + O\left(\sqrt{T\log T}\log\left(\frac{L^2\beta_T \log T}{\lambda\Gamma^2}\right)\right) .$$

**Lemma 9.** *Under Assumption 1, with the choice of $\gamma = \frac{1}{t^2}$ as in Eqn. (21), for the free parameter $0 < \Gamma < 1$, the term $F_4$ for LinIMED-3 satisfies:*

$$F_4 \leq T\Gamma + O(1) .$$

## B.2  PROOF OF LEMMA 6

*Proof.* We first partition the analysis into the cases $\hat{A}_t \neq A_t$ and $\hat{A}_t = A_t$ as follows:

$$F_1 = \mathbb{E}\sum_{t=1}^{T}\Delta_t \cdot \mathbb{1}\{B_t, C_t, D_t\}$$

$$= \mathbb{E}\sum_{t=1}^{T}\Delta_t \cdot \mathbb{1}\{B_t, C_t, D_t\} \cdot \mathbb{1}\left\{\hat{A}_t \neq A_t\right\} + \mathbb{E}\sum_{t=1}^{T}\Delta_t \cdot \mathbb{1}\{B_t, C_t, D_t\} \cdot \mathbb{1}\left\{\hat{A}_t = A_t\right\}$$

**Case 1:** If $\hat{A}_t \neq A_t$, this means that the index of $A_t$ is $I_{t,A_t} = \frac{\hat{\Delta}_{t,A_t}^2}{\beta_{t-1}\|X_t\|_{V_{t-1}^{-1}}^2} + \log\frac{1}{\beta_{t-1}\|X_t\|_{V_{t-1}^{-1}}^2}$.

Using the fact that $I_{t,A_t} \leq I_{t,\hat{A}_t}$ we have:

$$I_{t,A_t} = \frac{\hat{\Delta}_{t,A_t}^2}{\beta_{t-1}\|X_t\|_{V_{t-1}^{-1}}^2} + \log\frac{1}{\beta_{t-1}\|X_t\|_{V_{t-1}^{-1}}^2}$$

$$\leq \log T \wedge \log\frac{1}{\beta_{t-1}\|\hat{X}_t\|_{V_{t-1}^{-1}}^2}$$

$$\leq \log T.$$

Therefore

$$\frac{\hat{\Delta}^2_{t,A_t}}{\beta_{t-1}\|X_t\|^2_{V^{-1}_{t-1}}} + \log \frac{1}{\beta_{t-1}\|X_t\|^2_{V^{-1}_{t-1}}} \leq \log T \ . \tag{22}$$

If $\|X_t\|^2_{V^{-1}_{t-1}} \geq \frac{\Delta^2_t}{\beta_{t-1}}$, using the same procedure to get from Eqn. (15) to Eqn. (16), one has:

$$\mathbb{E}\sum_{t=1}^{T} \Delta_t \cdot \mathbb{1}\{B_t, C_t, D_t\} \cdot \mathbb{1}\left\{\hat{A}_t \neq A_t\right\} \cdot \mathbb{1}\left\{\|X_t\|^2_{V^{-1}_{t-1}} \geq \frac{\Delta^2_t}{\beta_{t-1}}\right\}$$

$$\leq \mathbb{E}\sum_{t=1}^{T} \Delta_t \cdot \mathbb{1}\left\{\|X_t\|^2_{V^{-1}_{t-1}} \geq \frac{\Delta^2_t}{\beta_{t-1}}\right\}$$

$$< T\Gamma + \frac{48d\beta_T}{\Gamma} \log\left(1 + \frac{8L^2\beta_T}{\lambda\Gamma^2}\right)$$

$$= T\Gamma + O\left(\frac{d\beta_T}{\Gamma}\log\left(1 + \frac{L^2\beta_T}{\lambda\Gamma^2}\right)\right).$$

Else if $\|X_t\|^2_{V^{-1}_{t-1}} < \frac{\Delta^2_t}{\beta_{t-1}}$, this implies that $\log\frac{1}{\beta_{t-1}\|X_t\|^2_{V^{-1}_{t-1}}} > \log\frac{1}{\Delta^2_t} \geq 0$. Then substituting the event $D_t := \{\hat{\Delta}_{t,A_t} \geq \varepsilon\}$ into Eqn. (22), we obtain

$$\frac{\varepsilon^2}{\beta_{t-1}\|X_t\|^2_{V^{-1}_{t-1}}} \leq \log T \ .$$

With $\sqrt{\lambda}S \geq 1$ we have $\beta_{t-1} \geq 1$ , then one has

$$\|X_t\|^2_{V^{-1}_{t-1}} \geq \frac{\varepsilon^2}{\beta_{t-1}\log T}.$$

Hence

$$\mathbb{E}\sum_{t=1}^{T} \Delta_t \cdot \mathbb{1}\left\{B_t, C_t, D_t, \hat{A}_t \neq A_t, \|X_t\|^2_{V^{-1}_{t-1}} < \frac{\Delta^2_t}{\beta_{t-1}}\right\}$$

$$\leq \mathbb{E}\sum_{t=1}^{T} \Delta_t \cdot \mathbb{1}\left\{\|X_t\|^2_{V^{-1}_{t-1}} \geq \frac{\varepsilon^2}{\beta_{t-1}\log T}\right\} \ .$$

With the choice of $\varepsilon = (1 - \frac{2}{\sqrt{\log T}})\Delta_t$, when $T \geq 149 > \exp(5)$, $\varepsilon > \frac{\Delta_t}{10}$, then performing the "peeling device" on $\Delta_t$ yields

$$\mathbb{E}\sum_{t=1}^{T} \Delta_t \cdot \mathbb{1}\left\{\|X_t\|^2_{V^{-1}_{t-1}} \geq \frac{\varepsilon^2}{\beta_{t-1}\log T}\right\} \cdot \mathbb{1}\{\Delta_t \geq \Gamma\}$$

$$\leq 149 + \mathbb{E}\sum_{t=1}^{T}\sum_{l=1}^{\lceil Q \rceil} \Delta_t \cdot \mathbb{1}\left\{2^{-l} < \Delta_t \leq 2^{-l+1}, \|X_t\|^2_{V^{-1}_{t-1}} \geq \frac{\varepsilon^2}{\beta_{t-1}\log T}\right\}$$

$$\leq O(1) + \mathbb{E}\sum_{l=1}^{\lceil Q \rceil} 2^{-l+1}\sum_{t=1}^{T} \mathbb{1}\left\{\|X_t\|^2_{V^{-1}_{t-1}} \geq \frac{\varepsilon^2}{\beta_{t-1}\log T}\right\}$$

$$\leq O(1) + \mathbb{E}\sum_{l=1}^{\lceil Q \rceil} 2^{-l+1}\sum_{t=1}^{T} \mathbb{1}\left\{\|X_t\|^2_{V^{-1}_{t-1}} \geq \frac{2^{-2l}}{100\beta_T\log T}\right\}$$

$$\leq O(1) + \mathbb{E} \sum_{l=1}^{\lceil Q \rceil} 2^{-l+1} \cdot 2^{2l} \cdot 600 d\beta_T (\log T) \log \left( 1 + \frac{2^{2l} \cdot 200 L^2 \beta_T \log T}{\lambda} \right)$$

$$\leq O(1) + \mathbb{E} \sum_{l=1}^{\lceil Q \rceil} 2^{l+1} \cdot 600 d\beta_T (\log T) \log \left( 1 + \frac{2^{2 \lceil Q \rceil} \cdot 200 L^2 \beta_T \log T}{\lambda} \right)$$

$$< O(1) + \frac{4800 d\beta_T \log T}{\Gamma} \log \left( 1 + \frac{800 L^2 \beta_T \log T}{\lambda \Gamma^2} \right).$$

Considering the event $\{\Delta_t < \Gamma\}$, we can upper bound the corresponding expectation as follows

$$\mathbb{E} \sum_{t=1}^{T} \Delta_t \cdot \mathbb{1} \left\{ \|X_t\|_{V_{t-1}^{-1}}^2 \geq \frac{\varepsilon^2}{\beta_{t-1} \log T} \right\} \cdot \mathbb{1} \left\{ \Delta_t < \Gamma \right\} \leq \mathbb{E} \sum_{t=1}^{T} \Delta_t \cdot \mathbb{1} \left\{ \Delta_t < \Gamma \right\} < T\Gamma.$$

Then

$$\mathbb{E} \sum_{t=1}^{T} \Delta_t \cdot \mathbb{1} \left\{ B_t, C_t, D_t, \hat{A}_t \neq A_t, \|X_t\|_{V_{t-1}^{-1}}^2 < \frac{\Delta_t^2}{\beta_{t-1}} \right\}$$

$$\leq \mathbb{E} \sum_{t=1}^{T} \Delta_t \cdot \mathbb{1} \left\{ \|X_t\|_{V_{t-1}^{-1}}^2 \geq \frac{\varepsilon^2}{\beta_{t-1} \log T} \right\}$$

$$= \mathbb{E} \sum_{t=1}^{T} \Delta_t \cdot \mathbb{1} \left\{ \|X_t\|_{V_{t-1}^{-1}}^2 \geq \frac{\varepsilon^2}{\beta_{t-1} \log T} \right\} \cdot \mathbb{1} \left\{ \Delta_t \geq \Gamma \right\}$$

$$+ \mathbb{E} \sum_{t=1}^{T} \Delta_t \cdot \mathbb{1} \left\{ \|X_t\|_{V_{t-1}^{-1}}^2 \geq \frac{\varepsilon^2}{\beta_{t-1} \log T} \right\} \cdot \mathbb{1} \left\{ \Delta_t < \Gamma \right\}$$

$$\leq O(1) + T\Gamma + \frac{4800 d\beta_T \log T}{\Gamma} \log \left( 1 + \frac{800 L^2 \beta_T \log T}{\lambda \Gamma^2} \right).$$

Hence

$$\mathbb{E} \sum_{t=1}^{T} \Delta_t \cdot \mathbb{1} \left\{ B_t, C_t, D_t, \hat{A}_t \neq A_t \right\}$$

$$= \mathbb{E} \sum_{t=1}^{T} \Delta_t \cdot \mathbb{1} \left\{ B_t, C_t, D_t, \hat{A}_t \neq A_t, \|X_t\|_{V_{t-1}^{-1}}^2 \geq \frac{\Delta_t^2}{\beta_{t-1}} \right\}$$

$$+ \mathbb{E} \sum_{t=1}^{T} \Delta_t \cdot \mathbb{1} \left\{ B_t, C_t, D_t, \hat{A}_t \neq A_t, \|X_t\|_{V_{t-1}^{-1}}^2 < \frac{\Delta_t^2}{\beta_{t-1}} \right\}$$

$$\leq T\Gamma + O \left( \frac{d\beta_T}{\Gamma} \log \left( 1 + \frac{L^2 \beta_T}{\lambda \Gamma^2} \right) \right) + O(1) + T\Gamma + \frac{4800 d\beta_T \log T}{\Gamma} \log \left( 1 + \frac{800 L^2 \beta_T \log T}{\lambda \Gamma^2} \right)$$

$$\leq T\Gamma + O \left( \frac{d\beta_T \log T}{\Gamma} \log \left( 1 + \frac{L^2 \beta_T \log T}{\lambda \Gamma^2} \right) \right).$$

**Case 2:** If $\hat{A}_t = A_t$, then from the event $C_t$ and the choice $\delta = \frac{\Delta_t}{\sqrt{\log T}}$ we have

$$\langle \hat{\theta}_{t-1} - \theta^*, X_t \rangle > \left( 1 - \frac{1}{\sqrt{\log T}} \right) \Delta_t.$$

Furthermore, using the definition of the event $B_t$, that implies that

$$\|X_t\|_{V_{t-1}^{-1}}^2 > \frac{(1 - \frac{1}{\sqrt{\log T}})^2 \Delta_t^2}{\beta_{t-1}}.$$

When $T > 8 > \exp(2)$, $(1 - \frac{1}{\sqrt{\log T}})^2 > \frac{1}{16}$, then similarily, we can bound this term by $O(\frac{d\beta_T}{\Gamma}) \log(1 + \frac{L^2 \beta_T}{\lambda \Gamma^2})$

Summarizing the two cases,

$$F_1 \leq O(1) + T\Gamma + O\left(\frac{d\beta_T \log T}{\Gamma}\right) \log\left(1 + \frac{L^2 \beta_T \log T}{\lambda \Gamma^2}\right)$$

$$\leq T\Gamma + O\left(\frac{d\beta_T \log T}{\Gamma}\right) \log\left(1 + \frac{L^2 \beta_T \log T}{\lambda \Gamma^2}\right).$$

□

### B.3 PROOF OF LEMMA 7

*Proof.* Recall that

$$F_2 = \mathbb{E} \sum_{t=1}^{T} \Delta_t \cdot \mathbb{1}\left\{B_t, C_t, \overline{D}_t\right\} .$$

From $C_t$ and $\overline{D}_t$, we derive that:

$$\langle \theta^*, a_t^* \rangle - \delta < \varepsilon + \langle \hat{\theta}_{t-1}, X_t \rangle.$$

With the choice $\delta = \frac{\Delta_t}{\sqrt{\log T}}, \varepsilon = (1 - \frac{2}{\sqrt{\log T}})\Delta_t$, we have

$$\langle \hat{\theta}_{t-1} - \theta^*, X_t \rangle > \frac{\Delta_t}{\sqrt{\log T}}. \tag{23}$$

Then using the definition of the event $B_t$ in Eqn. (23) yields

$$\|X_t\|^2_{V_{t-1}^{-1}} \geq \frac{\Delta_t^2}{\beta_{t-1} \log T}.$$

Using a similar procedure as in that from Eqn. (15) to Eqn. (16), we can upper bound $F_2$ by

$$F_2 \leq T\Gamma + O\left(\frac{d\beta_T \log T}{\Gamma}\right) \log\left(1 + \frac{L^2 \beta_T \log T}{\lambda \Gamma^2}\right).$$

□

### B.4 PROOF OF LEMMA 8

*Proof.* From the event $\overline{C}_t$, which is $\max_{b \in \mathcal{A}_t} \langle \hat{\theta}_{t-1}, b \rangle \leq \langle \theta^*, x_t^* \rangle - \delta$, the index of the best arm at time $t$ can be upper bounded as:

$$I_{t,a_t^*} \leq \frac{(\langle \theta^*, x_t^* \rangle - \delta - \langle \hat{\theta}_{t-1}, x_t^* \rangle)^2}{\beta_{t-1} \|x_t^*\|^2_{V_{t-1}^{-1}}} + \log \frac{1}{\beta_{t-1} \|x_t^*\|^2_{V_{t-1}^{-1}}}.$$

**Case 1:** If $\hat{A}_t \neq A_t$, then we have

$$I_{t,a_t^*} \geq I_{t,A_t} \geq \log \frac{1}{\beta_{t-1} \|X_t\|^2_{V_{t-1}^{-1}}}.$$

Suppose $\frac{q+1}{2}\delta \leq \langle \theta^*, x_t^* \rangle - \langle \hat{\theta}_{t-1}, x_t^* \rangle < \frac{q+2}{2}\delta$ for $q \in \mathbb{N}$, then one has

$$\log \frac{1}{\beta_{t-1}\|X_t\|^2_{V_{t-1}^{-1}}} \leq \frac{q^2 \delta^2}{4\beta_{t-1}\|x_t^*\|^2_{V_{t-1}^{-1}}} + \log \frac{1}{\beta_{t-1}\|x_t^*\|^2_{V_{t-1}^{-1}}}. \tag{24}$$

On the other hand, on the event $B_t$,

$$\|x_t^*\|_{V_{t-1}^{-1}} \geq \frac{(q+1)\delta}{2\sqrt{\beta_{t-1}}}. \tag{25}$$

If $\|X_t\|_{V_{t-1}^{-1}}^2 \geq \frac{\Delta_t^2}{\beta_{t-1}}$, using the same procedure from Eqn. (15) to Eqn. (16), one has:

$$\mathbb{E}\sum_{t=1}^T \Delta_t \cdot \mathbb{1}\left\{B_t, \overline{C}_t\right\} \cdot \mathbb{1}\left\{\hat{A}_t \neq A_t\right\} \cdot \mathbb{1}\left\{\|X_t\|_{V_{t-1}^{-1}}^2 \geq \frac{\Delta_t^2}{\beta_{t-1}}\right\}$$

$$\leq \mathbb{E}\sum_{t=1}^T \Delta_t \cdot \mathbb{1}\left\{\|X_t\|_{V_{t-1}^{-1}}^2 \geq \frac{\Delta_t^2}{\beta_{t-1}}\right\}$$

$$< T\Gamma + \frac{48d\beta_T}{\Gamma}\log\left(1 + \frac{8L^2\beta_T}{\lambda\Gamma^2}\right)$$

$$= T\Gamma + O\left(\frac{d\beta_T}{\Gamma}\log\left(1 + \frac{L^2\beta_T}{\lambda\Gamma^2}\right)\right).$$

Else if $\|X_t\|_{V_{t-1}^{-1}}^2 < \frac{\Delta_t^2}{\beta_{t-1}}$, this implies that $\log\frac{1}{\beta_{t-1}\|X_t\|_{V_{t-1}^{-1}}^2} > \log\frac{1}{\Delta_t^2} \geq 0$. Then combining Eqn. (24) and Eqn. (25) implies that

$$\|X_t\|_{V_{t-1}^{-1}}^2 \geq \frac{(q+1)^2\delta^2}{4\beta_{t-1}}\exp\left(-\frac{q^2}{(q+1)^2}\right).$$

Then using the same procedure to get from Eqn. (19) to Eqn. (20), we have

$$\sum_{t=1}^T \Delta_t \cdot \mathbb{1}\left\{B_t, \overline{C}_t\right\} \cdot \mathbb{1}\left\{\|X_t\|_{V_{t-1}^{-1}}^2 < \frac{\Delta_t^2}{\beta_{t-1}}, \hat{A}_t \neq A_t\right\}$$

$$< T\Gamma + O\left(\frac{d\beta_T \log T}{\Gamma}\log\left(1 + \frac{L^2\beta_T \log T}{\lambda\Gamma^2}\right)\right). \tag{26}$$

**Case 2:** $\hat{A}_t = A_t$. If $\|X_t\|_{V_{t-1}^{-1}}^2 \geq \frac{\Delta_t^2}{\beta_{t-1}}$, using the same procedure to get from Eqn. (15) to Eqn. (16), one has:

$$\mathbb{E}\sum_{t=1}^T \Delta_t \cdot \mathbb{1}\left\{B_t, \overline{C}_t\right\} \cdot \mathbb{1}\left\{\hat{A}_t = A_t\right\} \cdot \mathbb{1}\left\{\|X_t\|_{V_{t-1}^{-1}}^2 \geq \frac{\Delta_t^2}{\beta_{t-1}}\right\}$$

$$\leq \mathbb{E}\sum_{t=1}^T \Delta_t \cdot \mathbb{1}\left\{\|X_t\|_{V_{t-1}^{-1}}^2 \geq \frac{\Delta_t^2}{\beta_{t-1}}\right\}$$

$$< T\Gamma + \frac{48d\beta_T}{\Gamma}\log\left(1 + \frac{8L^2\beta_T}{\lambda\Gamma^2}\right)$$

$$= T\Gamma + O\left(\frac{d\beta_T}{\Gamma}\log\left(1 + \frac{L^2\beta_T}{\lambda\Gamma^2}\right)\right).$$

Else $\|X_t\|_{V_{t-1}^{-1}}^2 < \frac{\Delta_t^2}{\beta_{t-1}}$ implies that $\log\frac{1}{\beta_{t-1}\|X_t\|_{V_{t-1}^{-1}}^2} > \log\frac{1}{\Delta_t^2} \geq 0$.

If $\log\frac{1}{\beta_{t-1}\|X_t\|_{V_{t-1}^{-1}}^2} < \log T$, then using the same procedure to get from Eqn. (24) to Eqn. (26), we have

$$\sum_{t=1}^T \Delta_t \cdot \mathbb{1}\left\{B_t, \overline{C}_t\right\} \cdot \mathbb{1}\left\{\|X_t\|_{V_{t-1}^{-1}}^2 < \frac{\Delta_t^2}{\beta_{t-1}}, \hat{A}_t = A_t, \log\frac{1}{\beta_{t-1}\|X_t\|_{V_{t-1}^{-1}}^2} < \log\frac{T}{\beta_{t-1}}\right\}$$

$$< T\Gamma + O\left(\frac{d\beta_T \log T}{\Gamma}\log\left(1 + \frac{L^2\beta_T \log T}{\lambda\Gamma^2}\right)\right).$$

If $\log \frac{1}{\beta_{t-1}\|X_t\|^2_{V^{-1}_{t-1}}} \geq \log T$, this means now the index of $A_t$ is $I_{t,A_t} = \log T$, by performing the "peeling device" such that $\frac{q+1}{2}\delta \leq \langle\theta^*, x^*_t\rangle - \langle\hat{\theta}_{t-1}, x^*_t\rangle < \frac{q+2}{2}\delta$ for $q \in \mathbb{N}$, we have

$$\log T \leq \frac{q^2\delta^2}{4\beta_{t-1}\|x^*_t\|^2_{V^{-1}_{t-1}}} + \log \frac{1}{\beta_{t-1}\|x^*_t\|^2_{V^{-1}_{t-1}}}. \tag{27}$$

On the other hand, using the definition of the event $B_t$,

$$\|x^*_t\|_{V^{-1}_{t-1}} \geq \frac{(q+1)\delta}{2\sqrt{\beta_{t-1}}}. \tag{28}$$

Combining Eqn. (27) and (28), we have

$$\delta \leq \frac{2\exp(\frac{q^2}{2(q+1)^2})}{(q+1)\sqrt{T}}.$$

Then with $\delta = \frac{\Delta_t}{\sqrt{\log T}}$, this implies that

$$\Delta_t \leq \frac{2\sqrt{\log T}\exp(\frac{q^2}{2(q+1)^2})}{(q+1)\sqrt{T}}.$$

On the other hand, from $\frac{q+1}{2}\delta \leq \sqrt{\beta_{t-1}}\|x^*_t\|_{V^{-1}_{t-1}} \leq \sqrt{\beta_{t-1}} \cdot \frac{L}{\sqrt{\lambda}}$, we have $q+1 \leq \frac{2L\sqrt{\beta_{t-1}\log T}}{\sqrt{\lambda}\Delta_t}$. Hence,

$$\sum_{t=1}^{T}\Delta_t \cdot \mathbb{1}\left\{B_t, \overline{C}_t\right\} \cdot \mathbb{1}\left\{\|X_t\|^2_{V^{-1}_{t-1}} < \frac{\Delta^2_t}{\beta_{t-1}}, \hat{A}_t = A_t, \log\frac{1}{\beta_{t-1}\|X_t\|^2_{V^{-1}_{t-1}}} \geq \log T, \Delta_t \geq \Gamma\right\}$$

$$\leq \mathbb{E}\sum_{q=1}^{\lfloor\frac{2L\sqrt{\beta_T\log T}}{\sqrt{\lambda}\Gamma}-1\rfloor}\sum_{t=1}^{T}\Delta_t \cdot \mathbb{1}\left\{\Delta_t \leq \frac{2\sqrt{\log T}\exp(\frac{q^2}{2(q+1)^2})}{(q+1)\sqrt{T}}\right\}$$

$$\leq \mathbb{E}\sum_{q=1}^{\lfloor\frac{2L\sqrt{\beta_T\log T}}{\sqrt{\lambda}\Gamma}-1\rfloor}\sum_{t=1}^{T}\frac{2\sqrt{\log T}\exp(\frac{q^2}{2(q+1)^2})}{(q+1)\sqrt{T}}$$

$$= \mathbb{E}\sum_{q=1}^{\lfloor\frac{2L\sqrt{\beta_T\log T}}{\sqrt{\lambda}\Gamma}-1\rfloor}\frac{2\sqrt{T\log T}\exp(\frac{q^2}{2(q+1)^2})}{q+1}$$

$$< \mathbb{E}\sum_{q=1}^{\lfloor\frac{2L\sqrt{\beta_T\log T}}{\sqrt{\lambda}\Gamma}-1\rfloor}\frac{2\sqrt{e}\sqrt{T\log T}}{q+1}$$

$$< 2\sqrt{e}\sqrt{T\log T}\log\left(\frac{2L\sqrt{\log T}}{\sqrt{\lambda}\Gamma}-1\right)$$

$$\leq O\left(\sqrt{T\log T}\log\left(\frac{L^2\beta_T\log T}{\lambda\Gamma^2}\right)\right).$$

Summarizing the two cases ($\hat{A}_t \neq A_t$ and $\hat{A}_t = A_t$), we see that $F_3$ is upper bounded by:

$$
\begin{aligned}
F_3 < &\ T\Gamma + O\left(\frac{d\beta_T}{\Gamma}\log\left(1 + \frac{L^2\beta_T}{\lambda\Gamma^2}\right)\right) + T\Gamma + O\left(\frac{d\beta_T\log T}{\Gamma}\log\left(1 + \frac{L^2\beta_T\log T}{\lambda\Gamma^2}\right)\right) \\
&+ T\Gamma + O\left(\frac{d\beta_T}{\Gamma}\log\left(1 + \frac{L^2\beta_T}{\lambda\Gamma^2}\right)\right) + T\Gamma + O\left(\frac{d\beta_T\log T}{\Gamma}\log\left(1 + \frac{L^2\beta_T\log T}{\lambda\Gamma^2}\right)\right) \\
&+ T\Gamma + O\left(\sqrt{T\beta_T\log T}\log\left(\frac{L^2\beta_T\log T}{\lambda\Gamma^2}\right)\right) \\
\leq &\ 5T\Gamma + O\left(\frac{d\beta_T\log T}{\Gamma}\log\left(1 + \frac{L^2\beta_T\log T}{\lambda\Gamma^2}\right)\right) + O\left(\sqrt{T\log T}\log\left(\frac{L^2\beta_T\log T}{\lambda\Gamma^2}\right)\right).
\end{aligned}
$$

$\square$

### B.5 Proof of Lemma 9

*Proof.* The proof of this case is straightforward by using Lemma 1 with the choice $\gamma = \frac{1}{t^2}$:

$$
\begin{aligned}
F_4 &= \mathbb{E}\sum_{t=1}^{T}\Delta_t \cdot \mathbb{1}\left\{\overline{B}_t\right\} \\
&= \mathbb{E}\sum_{t=1}^{T}\Delta_t \cdot \mathbb{1}\left\{\overline{B}_t, \Delta_t < \Gamma\right\} + \mathbb{E}\sum_{t=1}^{T}\Delta_t \cdot \mathbb{1}\left\{\overline{B}_t, \Delta_t \geq \Gamma\right\} \\
&< T\Gamma + \mathbb{E}\sum_{t=1}^{T}\sum_{l=1}^{\lceil Q \rceil}\Delta_t \cdot \mathbb{1}\left\{\overline{B}_t, 2^{-l} < \Delta_t \leq 2^{-l+1}\right\} \\
&\leq T\Gamma + \mathbb{E}\sum_{t=1}^{T}\sum_{l=1}^{\lceil Q \rceil}2^{-l+1} \cdot \mathbb{1}\left\{\overline{B}_t\right\} \\
&\leq T\Gamma + \sum_{l=1}^{\lceil Q \rceil}2^{-l+1}\sum_{t=1}^{T}\mathbb{P}\left\{\overline{B}_t\right\} \\
&= T\Gamma + \sum_{l=1}^{\lceil Q \rceil}2^{-l+1} \cdot \frac{\pi^2}{6} \\
&< T\Gamma + (2-\Gamma) \cdot \frac{\pi^2}{6} \\
&< T\Gamma + \frac{\pi^2}{3} \\
&= T\Gamma + O(1).
\end{aligned}
$$

$\square$

### B.6 Proof of Theorem 2

*Proof.* Combining Lemmas 6, 7, 8 and 9, with the choices of $\gamma$ and $\Gamma$ as in Eqn. (21), the regret of LinIMED-2 is bounded as follows:

$$
\begin{aligned}
R_T &= F_1 + F_2 + F_3 + F_4 \\
&\leq T\Gamma + O\left(\frac{d\beta_T\log T}{\Gamma}\right)\log\left(1 + \frac{L^2\beta_T\log T}{\lambda\Gamma^2}\right) + T\Gamma + O\left(\frac{d\beta_T\log T}{\Gamma}\right)\log\left(1 + \frac{L^2\beta_T\log T}{\lambda\Gamma^2}\right) \\
&\quad + 5T\Gamma + O\left(\frac{d\beta_T\log T}{\Gamma}\log\left(1 + \frac{L^2\beta_T\log T}{\lambda\Gamma^2}\right)\right) + O\left(\sqrt{T\log T}\log\left(\frac{L^2\beta_T\log T}{\lambda\Gamma^2}\right)\right)
\end{aligned}
$$

$$+ T\Gamma + O(1)$$

$$\leq 8T\Gamma + O\left(\frac{d\beta_T \log T}{\Gamma} \log\left(1 + \frac{L^2 \beta_T \log T}{\lambda \Gamma^2}\right)\right) + O\left(\sqrt{T \log T} \log\left(\frac{L^2 \beta_T \log T}{\lambda \Gamma^2}\right)\right)$$

$$= 8\sqrt{dT\beta_T} \log T + O\left(\sqrt{dT\beta_T} \log\left(1 + \frac{TL^2}{\lambda d \log T}\right)\right) + O\left(\sqrt{T \log T} \log\left(\frac{TL^2}{\lambda d \log T}\right)\right)$$

$$= 8d\sqrt{T} \log^{\frac{3}{2}} T + O\left(d\sqrt{T} \log^{\frac{3}{2}} T\right) + O\left(\sqrt{T} \log^{\frac{3}{2}} T\right)$$

$$\leq O\left(d\sqrt{T} \log^{\frac{3}{2}} T\right).$$

$\square$

## C  PROOF OF THE REGRET BOUND FOR LINIMED-3 (PROOF OF THEOREM 3)

First we define $a_t^*$ as the best arm in time step $t$ such that $a_t^* = \arg\max_{a \in \mathcal{A}_t} \langle \theta^*, x_{t,a} \rangle$, and use $x_t^* := x_{t,a_t^*}$ denote its corresponding context. Define $\hat{A}_t := \arg\max_{a \in \mathcal{A}_t} \text{UCB}_t(a)$. Let $\Delta_t := \langle \theta^*, x_t^* \rangle - \langle \theta^*, X_t \rangle$ denote the regret in time $t$. Define the following events:

$$B_t' := \left\{ \|\hat{\theta}_{t-1} - \theta^*\|_{V_{t-1}} \leq \sqrt{\beta_{t-1}(\gamma)} \right\}, \quad D_t' := \left\{ \hat{\Delta}_{t,A_t} > \varepsilon \right\}.$$

where $\varepsilon$ is a free parameter set to be $\varepsilon = \frac{\Delta_t}{3}$ in this proof sketch.

Then the expected regret $R_T = \mathbb{E} \sum_{t=1}^{T} \Delta_t$ can be partitioned by events $B_t', D_t'$ such that:

$$R_T = \underbrace{\mathbb{E} \sum_{t=1}^{T} \Delta_t \cdot \mathbb{1}\left\{B_t', D_t'\right\}}_{=:F_1} + \underbrace{\mathbb{E} \sum_{t=1}^{T} \Delta_t \cdot \mathbb{1}\left\{B_t, \overline{D_t'}\right\}}_{=:F_2} + \underbrace{\mathbb{E} \sum_{t=1}^{T} \Delta_t \cdot \mathbb{1}\left\{\overline{B_t'}\right\}}_{=:F_3}.$$

**For the $F_1$ case:**
From $D_t'$ we know $A_t \neq \hat{A}_t$, therefore

$$I_{t,A_t} = \frac{\hat{\Delta}_{t,A_t}^2}{\beta_{t-1} \|X_t\|_{V_{t-1}^{-1}}^2} + \log \frac{1}{\beta_{t-1} \|X_t\|_{V_{t-1}^{-1}}^2} . \tag{29}$$

From $D_t'$ and $I_{t,A_t} \leq I_{t,\hat{A}_t} \leq \log \frac{C}{\max_{a \in \mathcal{A}_t} \hat{\Delta}_{t,a}^2}$, we have

$$I_{t,A_t} < \log \frac{C}{\varepsilon^2} . \tag{30}$$

Combining Eqn. (29) and Eqn. (30),

$$\frac{\hat{\Delta}_{t,A_t}^2}{\beta_{t-1} \|X_t\|_{V_{t-1}^{-1}}^2} + \log \frac{1}{\beta_{t-1} \|X_t\|_{V_{t-1}^{-1}}^2} < \log \frac{C}{\varepsilon^2} .$$

Then

$$\frac{\hat{\Delta}_{t,A_t}^2}{\beta_{t-1} \|X_t\|_{V_{t-1}^{-1}}^2} < \log \beta_{t-1} \|X_t\|_{V_{t-1}^{-1}}^2 \cdot \frac{C}{\varepsilon^2} . \tag{31}$$

If $\|X_t\|_{V_{t-1}^{-1}}^2 \geq \frac{\Delta_t^2}{\beta_{t-1}}$, using the same procedure from Eqn. (15) to Eqn. (16), one has:

$$
\mathbb{E} \sum_{t=1}^{T} \Delta_t \cdot \mathbb{1}\{B_t', D_t'\} \cdot \mathbb{1}\left\{ \|X_t\|_{V_{t-1}^{-1}}^2 \geq \frac{\Delta_t^2}{\beta_{t-1}} \right\}
$$

$$
\leq \mathbb{E} \sum_{t=1}^{T} \Delta_t \cdot \mathbb{1}\left\{ \|X_t\|_{V_{t-1}^{-1}}^2 \geq \frac{\Delta_t^2}{\beta_{t-1}} \right\}
$$

$$
< T\Gamma + \frac{48 d \beta_T}{\Gamma} \log\left( 1 + \frac{8 L^2 \beta_T}{\lambda \Gamma^2} \right)
$$

$$
= T\Gamma + O\left( \frac{d \beta_T}{\Gamma} \log\left( 1 + \frac{L^2 \beta_T}{\lambda \Gamma^2} \right) \right).
$$

Else $\|X_t\|_{V_{t-1}^{-1}}^2 < \frac{\Delta_t^2}{\beta_{t-1}}$, this implies that $\beta_{t-1}\|X_t\|_{V_{t-1}^{-1}}^2 < \Delta_t^2$, plug this into Eqn. (31) and with the choice of $\varepsilon = \frac{\Delta_t}{3}$ and $D_t'$, we have

$$
\frac{\Delta_t^2}{9 \beta_{t-1} \|X_t\|_{V_{t-1}^{-1}}^2} < \log(9C) .
$$

Since $C \geq 1$ is a constant, then

$$
\|X_t\|_{V_{t-1}^{-1}}^2 > \frac{\Delta_t^2}{9 \beta_{t-1} \log(9C)} .
$$

Using the same procedure from Eqn. (15) to Eqn. (16), one has:

$$
\mathbb{E} \sum_{t=1}^{T} \Delta_t \cdot \mathbb{1}\{B_t', D_t'\} \cdot \mathbb{1}\left\{ \|X_t\|_{V_{t-1}^{-1}}^2 < \frac{\Delta_t^2}{\beta_{t-1}} \right\}
$$

$$
\leq \mathbb{E} \sum_{t=1}^{T} \Delta_t \cdot \mathbb{1}\left\{ \|X_t\|_{V_{t-1}^{-1}}^2 > \frac{\Delta_t^2}{9 \beta_{t-1} \log(9C)} \right\}
$$

$$
< T\Gamma + O\left( \frac{d \beta_T \log C}{\Gamma} \log\left( 1 + \frac{L^2 \beta_T \log C}{\lambda \Gamma^2} \right) \right).
$$

Hence

$$
F_1 < 2T\Gamma + O\left( \frac{d \beta_T \log C}{\Gamma} \log\left( 1 + \frac{L^2 \beta_T \log C}{\lambda \Gamma^2} \right) \right). \tag{32}
$$

**For the $F_2$ case:** Since the event $B_t'$ holds,

$$
\max_{a \in \mathcal{A}_t} \mathrm{UCB}_t(a) \geq \mathrm{UCB}_t(a_t^*) = \langle \hat{\theta}_{t-1}, x_t^* \rangle + \sqrt{\beta_{t-1}} \|x_t^*\|_{V_{t-1}^{-1}} \geq \langle \theta^*, x_t^* \rangle \tag{33}
$$

On the other hand, from $\overline{D_t'}$ we have

$$
\max_{a \in \mathcal{A}_t} \mathrm{UCB}_t(a) \leq \mathrm{UCB}_t(A_t) + \varepsilon = \langle \hat{\theta}_{t-1}, X_t \rangle + \sqrt{\beta_{t-1}} \|X_t\|_{V_{t-1}^{-1}} + \varepsilon . \tag{34}
$$

Combining Eqn. (33) and Eqn. (34),

$$
\langle \theta^*, x_t^* \rangle \leq \langle \hat{\theta}_{t-1}, X_t \rangle + \sqrt{\beta_{t-1}} \|X_t\|_{V_{t-1}^{-1}} + \varepsilon .
$$

Hence

$$
\Delta_t - \varepsilon \leq \langle \hat{\theta}_{t-1} - \theta^*, X_t \rangle + \sqrt{\beta_{t-1}} \|X_t\|_{V_{t-1}^{-1}} .
$$

Then with $\varepsilon = \frac{\Delta_t}{3}$ and $B_t'$, we have

$$
\frac{2}{3} \Delta_t \leq 2\sqrt{\beta_{t-1}} \|X_t\|_{V_{t-1}^{-1}} ,
$$

therefore

$$\|X_t\|_{V_{t-1}^{-1}}^2 > \frac{\Delta_t^2}{9\beta_{t-1}} \ .$$

Using the same procedure from Eqn. (15) to Eqn. (16), one has:

$$F_2 < T\Gamma + O\left(\frac{d\beta_T}{\Gamma} \log\left(1 + \frac{L^2\beta_T}{\lambda\Gamma^2}\right)\right) . \tag{35}$$

**For the $F_3$ case:**
using Lemma 1 with the choice $\gamma = \frac{1}{t^2}$:

$$
\begin{aligned}
F_3 &= \mathbb{E}\sum_{t=1}^{T}\Delta_t \cdot \mathbb{1}\left\{\overline{B_t'}\right\} \\
&= \mathbb{E}\sum_{t=1}^{T}\Delta_t \cdot \mathbb{1}\left\{\overline{B_t'}, \Delta_t < \Gamma\right\} + \mathbb{E}\sum_{t=1}^{T}\Delta_t \cdot \mathbb{1}\left\{\overline{B_t'}, \Delta_t \geq \Gamma\right\} \\
&< T\Gamma + \mathbb{E}\sum_{t=1}^{T}\sum_{l=1}^{\lceil Q\rceil}\Delta_t \cdot \mathbb{1}\left\{\overline{B_t'}, 2^{-l} < \Delta_t \leq 2^{-l+1}\right\} \\
&\leq T\Gamma + \mathbb{E}\sum_{t=1}^{T}\sum_{l=1}^{\lceil Q\rceil}2^{-l+1} \cdot \mathbb{1}\left\{\overline{B_t'}\right\} \\
&\leq T\Gamma + \sum_{l=1}^{\lceil Q\rceil}2^{-l+1}\sum_{t=1}^{T}\mathbb{P}\left\{\overline{B_t'}\right\} \\
&= T\Gamma + \sum_{l=1}^{\lceil Q\rceil}2^{-l+1} \cdot \frac{\pi^2}{6} \\
&< T\Gamma + (2 - \Gamma) \cdot \frac{\pi^2}{6} \\
&< T\Gamma + \frac{\pi^2}{3} \\
&= T\Gamma + O(1) . 
\end{aligned}
\tag{36}
$$

### C.1 Proof of Theorem 3

*Proof.* Combining Eqn. (32), (35), (36) with the choices of $\gamma = \frac{1}{t^2}$ and $\Gamma = \frac{\beta_T}{\sqrt{T}}$ and $C \geq 1$ is a constant, the regret of LinIMED-3 is bounded as follows:

$$
\begin{aligned}
R_T &= F_1 + F_2 + F_3 + F_4 \\
&< 4T\Gamma + O\left(\frac{d\beta_T\log C}{\Gamma}\log\left(1 + \frac{L^2\beta_T\log C}{\lambda\Gamma^2}\right)\right) + O\left(\frac{d\beta_T}{\Gamma}\log\left(1 + \frac{L^2\beta_T}{\lambda\Gamma^2}\right)\right) + O(1) \\
&< O\left(d\sqrt{T}\log C\log\left(1 + \frac{L^2 T\log C}{\lambda}\right)\right) \\
&= O\left(d\sqrt{T}\log(T)\right) .
\end{aligned}
$$

$\square$

## D Hyperparameter tuning in our empirical study

### D.1 Synthetic Dataset

The below tables are the empirical results while tuning the hyperparameter $\alpha$ (scale of the confidence width) for fixed $T = 100$.

| Method | LinUCB | | | LinTS | | | LinIMED-1 | | | LinIMED-2 | | | LinIMED-3 ($C = 30$) | | |
|---|---|---|---|---|---|---|---|---|---|---|---|---|---|---|---|
| $\alpha$ | 0.6 | 0.7 | 0.8 | 0.3 | 0.4 | 0.5 | 0.4 | 0.5 | 0.6 | 0.4 | 0.5 | 0.6 | 0.3 | 0.4 | 0.5 |
| Regret | 3.38 | 3.28 | 3.37 | 3.82 | 3.28 | 3.99 | 3.23 | 3.16 | 3.38 | 3.23 | 3.18 | 3.23 | 3.19 | 3.01 | 3.28 |

Table 2: Tuning $\alpha$ when $K = 10, d = 20$

| Method | LinUCB | | | LinTS | | | LinIMED-1 | | | LinIMED-2 | | | LinIMED-3 ($C = 30$) | | |
|---|---|---|---|---|---|---|---|---|---|---|---|---|---|---|---|
| $\alpha$ | 0.9 | 1.0 | 1.1 | 0.3 | 0.4 | 0.5 | 0.3 | 0.4 | 0.5 | 0.5 | 0.6 | 0.7 | 0.4 | 0.5 | 0.6 |
| Regret | 3.74 | 3.63 | 3.64 | 4.39 | 3.39 | 4.36 | 3.66 | 3.50 | 3.75 | 3.535 | 3.533 | 3.945 | 3.44 | 3.36 | 3.88 |

Table 3: Tuning $\alpha$ when $K = 100, d = 20$

| Method | LinUCB | | | LinTS | | | LinIMED-1 | | | LinIMED-2 | | | LinIMED-3 ($C = 30$) | | |
|---|---|---|---|---|---|---|---|---|---|---|---|---|---|---|---|
| $\alpha$ | 0.5 | 0.6 | 0.7 | 0 | 0.1 | 0.2 | 0.5 | 0.6 | 0.7 | 0.4 | 0.5 | 0.6 | 0.4 | 0.5 | 0.6 |
| Regret | 3.30 | 3.29 | 3.34 | 7.00 | 2.52 | 2.62 | 3.16 | 3.07 | 3.41 | 3.33 | 3.17 | 3.26 | 3.02 | 3.00 | 3.53 |

Table 4: Tuning $\alpha$ when $K = 1000, d = 20$

| Method | LinUCB | | | LinTS | | | LinIMED-1 | | | LinIMED-2 | | | LinIMED-3 ($C = 30$) | | |
|---|---|---|---|---|---|---|---|---|---|---|---|---|---|---|---|
| $\alpha$ | 0.9 | 1 | 1.1 | 0.1 | 0.2 | 0.3 | 0.4 | 0.5 | 0.6 | 0.1 | 0.2 | 0.3 | 0.3 | 0.4 | 0.5 |
| Regret | 3.29 | 3.28 | 3.36 | 3.68 | 3.26 | 3.92 | 3.16 | 3.11 | 3.46 | 4.51 | 3.18 | 3.28 | 3.01 | 2.99 | 3.45 |

Table 5: Tuning $\alpha$ when $K = 10, d = 50$

| Method | LinUCB | | | LinTS | | | LinIMED-1 | | | LinIMED-2 | | | LinIMED-3 ($C = 30$) | | |
|---|---|---|---|---|---|---|---|---|---|---|---|---|---|---|---|
| $\alpha$ | 0.3 | 0.4 | 0.5 | 0 | 0.1 | 0.2 | 0.1 | 0.2 | 0.3 | 0.1 | 0.2 | 0.3 | 0.2 | 0.3 | 0.4 |
| Regret | 3.33 | 3.23 | 3.31 | 11.0 | 3.98 | 3.36 | 3.61 | 3.21 | 3.25 | 4.40 | 3.18 | 3.26 | 3.12 | 3.00 | 3.35 |

Table 6: Tuning $\alpha$ when $K = 10, d = 100$

We run these algorithms on the same dataset with different choices of $\alpha$, we choose the best $\alpha$ with the corresponding least regret.

## D.2 MovieLens Dataset

The below tables are the empirical results while tuning the hyper-parameter $\alpha$ (scale of the confidence width) for fixed $T = 100$.

| Method | LinUCB | | | LinTS | | | LinIMED-1 | | | LinIMED-2 | | | LinIMED-3 ($C = 30$) | | |
|---|---|---|---|---|---|---|---|---|---|---|---|---|---|---|---|
| $\alpha$ | 0.6 | 0.7 | 0.8 | 0.05 | 0.1 | 0.15 | 0.1 | 0.15 | 0.2 | 0.1 | 0.15 | 0.2 | 0.1 | 0.15 | 0.2 |
| CTR | 0.706 | 0.759 | 0.756 | 0.670 | 0.712 | 0.696 | 0.740 | 0.759 | 0.699 | 0.744 | 0.759 | 0.699 | 0.747 | 0.776 | 0.714 |

Table 7: Tuning $\alpha$ when $K = 50$

| Method | LinUCB | | | LinTS | | | LinIMED-1 | | | LinIMED-2 | | | LinIMED-3 ($C = 30$) | | |
|---|---|---|---|---|---|---|---|---|---|---|---|---|---|---|---|
| $\alpha$ | 0.8 | 0.9 | 1.0 | 0 | 0.05 | 0.1 | 0.05 | 0.1 | 0.15 | 0.05 | 0.1 | 0.15 | 0.05 | 0.1 | 0.15 |
| CTR | 0.538 | 0.584 | 0.531 | 0.395 | 0.520 | 0.384 | 0.559 | 0.612 | 0.525 | 0.464 | 0.619 | 0.535 | 0.453 | 0.598 | 0.537 |

Table 8: Tuning $\alpha$ when $K = 100$

| Method | LinUCB | | | LinTS | | | LinIMED-1 | | | LinIMED-2 | | | LinIMED-3 ($C = 30$) | | |
|---|---|---|---|---|---|---|---|---|---|---|---|---|---|---|---|
| $\alpha$ | 0.9 | 1.0 | 1.1 | 0 | 0.05 | 0.1 | 0.05 | 0.1 | 0.15 | 0.05 | 0.1 | 0.15 | 0.05 | 0.1 | 0.15 |
| CTR | 0.509 | 0.553 | 0.53 | 0.319 | 0.412 | 0.411 | 0.457 | 0.523 | 0.437 | 0.461 | 0.526 | 0.437 | 0.444 | 0.548 | 0.490 |

Table 9: Tuning $\alpha$ when $K = 500$

We run these algorithms on the same dataset with different choices of $\alpha$, we choose the best $\alpha$ with the corresponding largest reward.