# OpenReview forum: "Linear Indexed Minimum Empirical Divergence Algorithms"
_ICLR.cc/2024/Conference — ICLR 2024 Conference Withdrawn Submission_

### Official Review · Reviewer_iSL5 · 2023-10-28

**Soundness:** 2 fair
**Presentation:** 2 fair
**Contribution:** 1 poor
**Rating:** 3
**Confidence:** 4

**Summary:**

The paper provides linear versions of the Indexed Minimum Empirical Divergence (IMED) algorithm, that aims to solve the stochastic linear bandit problem. The paper introduces three versions of the linear IMED algorithm with different updating rules, and proves regret upper bounds. The paper provides proofs and some numerical studies.

**Strengths:**

The paper learns from extension of Thompson sampling algorithm to linear bandits, adopts that idea, and uses the idea to extend IMED algorithm to linear bandit problems. Regret bounds that match those of LinUCB with OFUL’s confidence bound are proved. This is a neat paper.

**Weaknesses:**

The paper's contribution should be insufficient for ICLR. The paper has done a decent application by extending to the linear setting for IMED algoritms. However, the algorithm design and supporting proofs are not novel, thereby having limited contribution toward the linear bandit literature.

The paper writes from page 5 to page 8 about their proof sketch. This part is not clearly written, and I am not sure what is the purpose of this part of the paper.

The paper's numerical study does not support the claims. The paper claims "Compared to OFUL, LinIMED works more efficiently" and "Compared to SupLinUCB, ..., our empirical performance is better." The numerical study is too incomprehensive to make such claims.

**Questions:**

Why do the authors believe that their numerical study can support their claims?

What is the purpose of page 5 to page 8 and what are the points that the authors want to make from this part?

---

### Official Review · Reviewer_FSvU · 2023-10-29

**Soundness:** 2 fair
**Presentation:** 2 fair
**Contribution:** 2 fair
**Rating:** 3
**Confidence:** 3

**Summary:**

The paper proposes novel algorithms for solving the stochastic linear bandit problem based on the principle of Indexed Minimum Empirical Divergence (IMED). Three variants of the same approach are proposed (namely, LinIMED 1-2-3), differing mainly on managing the exploration-exploitation trade-off. Theoretical guarantees and empirical evaluations of the three approaches are presented.

**Strengths:**

This paper provides new solutions to the well-known stochastic linear bandit setting, building on top of a principle different from Thompson sampling or pure optimism (even though many elements are in common), possibly leading to improved empirical performances. Moreover, proving the theoretical guarantees of the proposed algorithms required a non-trivial technical effort.

**Weaknesses:**

1) Overall, I felt that the presentation was not always clear. I’ll provide some examples:
    - I would have divided the pseudocode for the three algorithms, avoiding reporting all of them in the same. An idea would be to just provide the pseudocode for one and the other in the appendix since they are very similar.
    - The exploration term of the decision index is sometimes reported with a positive sign and a fraction inside the logarithm, and other times with a negative sign and a flipped logarithm argument.
    - I feel that there’s a contradiction when you say, first, that “… we replace the optimized KL-divergence measure in IMED in Eqn. (1) with the squared gap between the sample mean of arm i and the arm with the maximum sample mean. This choice simplifies our analysis and does not adversely affect the regret bound.”, but later, in the conclusions, you state “In the future, a fruitful direction of research is to analyze the effect of directly using the KL-divergence in Line 14 of Algorithm 1 instead of the estimated squared gap; we believe that in this case, the analysis would be more challenging, but the theoretical and empirical performances might be superior to our three LinIMED algorithms.”

2) Referring to the last point of the previous list, I think that this is a crucial thing about this paper: you state that you are providing “novel linear versions of the IMED algorithm” but, in LinIMED, I cannot see any divergence, but rather a rough proxy defined by the empirical gap. I think that, in terms of performance, this would not drastically affect your results, while from a technical point of view, this would drastically harden the difficulty. However, from the IMED counterpart for linear bandits, one would expect to see the divergence principle in action. At the same time, here, I’m not sure we can really talk of “Indexed Minimum Empirical Divergence” in a strict sense. I feel that addressing this point is a crucial task to strengthen the value of your work.

3) Looking at the plots of your experimental campaign, it’s hard to tell, for most of the time, if an algorithm is better than another. The fact is that most of the time the uncertainty bands around the average cumulative regrets are very much overlapped, leading us to no conclusion on whether an algorithm is effectively better than another in that experiment. I am trying to understand if this is a matter of how you computed that uncertainty or if there aren’t really any statistically sound conclusions that can be drawn from there. I would ask you to clarify this point.

**Questions:**

Even though theorem proving is non-trivial from a technical point of view, I think that most of the key ideas there came from well-known results in this literature (see, e.g., proofs of theoretical guarantees for LinUCB). Can you highlight what are, in your opinion, the main technical novelties in the passages of your proofs?

I am willing to revise my score if the authors can justify not using the true divergence instead of the estimated gap and if they can provide a satisfying answer to the previous question.

---

### Official Review · Reviewer_z91R · 2023-10-31

**Soundness:** 3 good
**Presentation:** 3 good
**Contribution:** 2 fair
**Rating:** 5
**Confidence:** 3

**Summary:**

This paper extends the Minimum Empirical Divergence algorithm to the linear bandit setting.  The authors provide three methods to do this and provide the regret guarantees for them. They show that the regret guarantees for the three algorithms are comparable to the other state-of-the-art algorithms for SLBs. They also show the efficacy of their method via empirical studies.

**Strengths:**

1) The work extends the MED framework to the linear bandit setting, closing the theoretical gap in the literature for SLBs.
2) The regret guarantees shown are optimal (up to log terms).
3) The numerical results justify their claims.

**Weaknesses:**

The main weakness of the work, in my opinion, lies in the contribution. I wish to know what were the main analytical changes faced in the extension of MED to LBs. It would be great if the authors provided some highlights on this in their final version because the main contribution of this work is theoretical. The experiments support their claims but do not drastically improve over LinUCB, which is the standard go-to for the SLB problem.

**Questions:**

1) "OFUL involves a bilinear optimization, which is computationally expensive unless the confidence ellipsoid is a convex hull of a finite set."Can the authors please comment on this in detail, since the bilinear optimization has a closed form and the computation necessary is linear in K, (and O(d^3))? Am I missing something?

2) Typo: T_i(t) = \sum_{s=1}^{t−1}{A_t = a} in Sec 3.1, t-->s, a-->i.
3) What were the main technical challenges that the authors faced while extending the analysis to the linear bandit setup? It would be nice to have a small subsection highlighting those.

---

### Official Review · Reviewer_PVPY · 2023-11-01

**Soundness:** 3 good
**Presentation:** 3 good
**Contribution:** 1 poor
**Rating:** 3
**Confidence:** 4

**Summary:**

This paper studies how to apply the IMED algorithm for semi-bounded multi-armed bandits to the (oblivious and finite-armed) linear bandits setting. As a result, the authors proposed the LinIMED algorithm based on the original IMED algorithm and the confidence ellipsoid in (Abbasi-Yadkori et al., 2011). The authors proved its regret bound which was $\widetilde O(d\sqrt T) $ omitting $\mathrm{poly}(\log T)$ factors. The authors also presented experimental results based on synthetic data to demonstrate the advantage of LinIMED upon existing algorithms for linear bandits including OFUL and linear Thompson sampling.

**Strengths:**

1. This is the first paper applying IMED to the linear bandits setting.
2. The authors proved the regret bound matching the minimax optimal regret bound for infinite-armed linear bandits up to polylog factors.
3. The authors presented experimental results for empirical comparison with existing algorithms.

**Weaknesses:**

1. The bound is not minimax optimal for finite-armed linear bandits. It has an $O(\sqrt d)$ from optimal algorithms such as SupLinUCB (Chu et al., 2011).
2. The most important assumption made by the IMED algorithm in (Honda and Takemura, 2015) was that the reward function could be semi-bounded, specifically it need not be bounded below. This distinguishes the IMED algorithm from the classical algorithms like (linear versions of) UCB. However, the authors made the assumption that the reward function was bounded.

**Questions:**

1. Did the authors make an assumption that $T \ge O(d)$?